# A Bi-metric Framework for Efficient Nearest Neighbor Search

## Abstract

We propose a new "bi-metric" framework for designing nearest neighbor data structures. Our framework assumes two dissimilarity functions: a *ground-truth* metric that is accurate but expensive to compute, and a *proxy* metric that is cheaper but less accurate. In both theory and practice, we show how to construct data structures using only the proxy metric such that the query procedure achieves the accuracy of the expensive metric, while only using a limited number of calls to both metrics. Our theoretical results instantiate this framework for two popular nearest neighbor search algorithms: DiskANN and Cover Tree. In both cases we show that, as long as the proxy metric used to construct the data structure approximates the ground-truth metric up to a bounded factor, our data structure achieves arbitrarily good approximation guarantees with respect to the ground-truth metric. On the empirical side, we apply the framework to the text retrieval problem with two dissimilarity functions evaluated by ML models with vastly different computational costs. We observe that for almost all the large data sets in the BEIR benchmark, our approach achieves a considerably better accuracy-efficiency tradeoff than the alternatives, such as retrieve-then-rerank.

## 1 Introduction

Similarity search is a versatile and popular approach to data retrieval. It assumes that the data items of interest (text passages, images, etc.) are equipped with a distance function, which for any pair of items estimates their similarity or dissimilarity. Then, given a "query" item, the goal is to return the data item that is most similar to the query. From the algorithmic perspective, this approach is formalized as the nearest neighbor search (NN) problem: given a set of $n$ points $P$ in a metric space $(X, D)$, build a data structure that, given any query point $q \in X$, returns $p \in P$ that minimizes $D(p, q)$. In many cases, the items are represented by high-dimensional feature vectors and $D$ is induced by the Euclidean distance between the vectors. In other cases, $D(p, q)$ is computed by a dedicated procedure given $p$ and $q$ (e.g., by a cross-encoder).

Over the last decade, mapping data items to feature vectors, or estimation of similarity between pairs of data items, is often done using ML models. (In the context of text retrieval, the first task is achieved by constructing bi-encoders (Karpukhin et al., 2020; Neelakantan et al., 2022; Gao et al., 2021b; Wang et al., 2024), while the second task uses cross-encoders (Gao et al., 2021a; Nogueira et al., 2020; Nogueira & Cho, 2020)). This creates efficiency bottlenecks, as high-accuracy models are often larger and slower, while cheaper models do not achieve the state-of-the-art accuracy. Furthermore, high-accuracy models are often proprietary and accessible only through a limited interface at a monetary cost. This motivates studying "the best of both worlds" solutions which utilize many types of models to achieve favorable tradeoffs between efficiency, accuracy and flexibility.

One popular method for combining multiple models is based on retrieve-then-rerank (Liu et al., 2009). It assumes two models: one model evaluating the metric $D$, which has high accuracy but is less efficient; and another model computing a "proxy" metric $d$, which is cheap but less accurate. The algorithm uses the second model ($d$) to retrieve a large (say, $k = 1000$) number of data items with the highest similarity to the query, and then uses the first model ($D$) to select the most similar items. The hyperparameter $k$ controls the tradeoff between the accuracy and efficiency. To improve the efficiency further, the retrieval of the top-$k$ items is typically accomplished using approximate

nearest neighbor data structures. Such data structures are constructed for the proxy metric $d$, so they remain stable even if the high-accuracy metric $D$ undergoes frequent updates.

Despite its popularity, the retrieve-then-rerank approach suffers from several issues:

1. The overall accuracy is limited by the accuracy of the cheaper model. To illustrate this phenomenon, suppose that $D$ defines the "true" distance, while $d$ only provides a "$C$-approximate" distance, i.e., that the values of $d$ and $D$ for the same pairs of items differ by at most a factor of $C > 1$. Then the re-ranking approach can only guarantee that the top reported item is a $C$-approximation, namely that its distance to the query is at most $C$ times the distance from the query to its true nearest neighbor according to $D$. This occurs because the first stage of the process, using the proxy $d$, might not retain the most relevant items.

2. Since the set of the top-$k$ items with respect to the more accurate model depends on the query, one needs to perform at least a linear scan over all $k$ data items retrieved using the proxy metric $d$. This computational cost can be reduced by decreasing $k$, but at the price of reducing the accuracy.

**Our results**  We show that, in both theory and practice, it is possible to combine cheap and expensive models to achieve approximate nearest neighbor data structures that inherit the accuracy of expensive models while significantly reducing the overall computational cost. Specifically, we propose a *bi-metric framework* for designing nearest neighbor data structures with the following properties: **(1)** The algorithm for creating the data structure uses only the proxy metric $d$, making it efficient to construct. **(2)** The algorithm for answering the nearest neighbor query leverages both models, but performs only a sub-linear number of evaluations of $d$ and $D$. **(3)** The data structure achieves the accuracy of the expensive model.

For a more formal description of the framework, see Preliminaries (Section 2).

The simplest approach to constructing algorithms that conform to our framework is to *construct the data structure using the proxy metric $d$, but answer queries using the accurate metric $D$*; we also propose more complex solutions with better performance. Our approach is quite general, and is applicable to any approximate nearest neighbor data structure for general metrics. Our *theoretical* study analyzes the simple approach when applied to two popular algorithms: DiskANN (Jayaram Subramanya et al., 2019) and Cover Tree (Beygelzimer et al., 2006), under natural assumptions about the intrinsic dimensionality of the data, as in Indyk & Xu (2023). Perhaps surprisingly, we show that despite the fact that only the proxy $d$ is used in the indexing stage, the query answering procedure essentially retains the accuracy of the ground truth metric $D$.

Formally, we show the following theorem statement. We use $\lambda_d$ to refer to the doubling dimension with respect to metric $d$ (a measure of intrinsic dimensionality, see Definition 2.2).

**Theorem 1.1** (Summary, see Theorems 3.3 and B.3). *Given a dataset $X$ of $n$ points, $\mathtt{Alg} \in \{DiskANN, Cover\ Tree\}$, and a fixed metric $d$, let $S_{\mathtt{Alg}}(n, \varepsilon, \lambda_d)$ and $Q_{\mathtt{Alg}}(\varepsilon, \lambda_d)$ denote the space and query complexity respectively of the standard datastructure for $\mathtt{Alg}$ which reports a $1 + \varepsilon$ nearest neighbor in $X$ for any query (all for a fixed metric $d$).*

*Consider two metrics $d$ and $D$ satisfying Equation 1. Then for any $\mathtt{Alg} \in \{DiskANN, Cover\ Tree\}$, we can build a corresponding datastructure $\mathcal{D}_{\mathtt{Alg}}$ on $X$ with the following properties:*

1. *When constructing $\mathcal{D}_{\mathtt{Alg}}$, we only access metric $d$,*
2. *The space used by $\mathcal{D}_{\mathtt{Alg}}$ can be bounded by $\tilde{O}(S_{\mathtt{Alg}}(n, \varepsilon/C, \lambda_d))$[1],*
3. *Given any query $q$, $\mathcal{D}_{\mathtt{Alg}}$ invokes $D$ at most $\tilde{O}(Q_{\mathtt{Alg}}(\varepsilon/C, \lambda_d))$ times,*
4. *$\mathcal{D}_{\mathtt{Alg}}$ returns a $1 + \varepsilon$ approximate nearest neighbor of $q$ in $X$ under metric $D$.*

The proof of the theorem crucially uses the properties of the underlying graph-based data structures. In Appendix F, we theoretically show that such a result is impossible to achieve for another popular family of nearest neighbor algorithms based on *locality sensitive hashing* (and other similar methods). Thus our work further highlights the *power* of graph-based methods, both theoretically and empirically.

To demonstrate the *practical* applicability of the bi-metric framework, we apply it to the text retrieval problem. Here, the data items are text passages, and the goal is to retrieve a passage from a large

---

[1]$\tilde{O}$ hides logarithm dependencies in the aspect ratio.

collection that is most relevant to a query passage. We instantiated our framework with the DiskANN algorithm. We use a lower-quality "bge-micro-v2" embedding model (AI, 2023) to define the metric $d$; the value of $d(p, q)$ is defined by the Euclidean distance between the embeddings of $p$ and $q$. The high-quality model $D$ is defined by one of the following two settings: **(1)** The *SFR-Embedding-Mistral* embedding model (Meng et al., 2024), where the metric is defined as the Euclidean distance between embeddings, and **(2)** The *Gemini-2.0-Flash* large language model. Here, we use the fact that graph-based algorithms for nearest neighbor search do not require the values $D(p, q)$ per se, but only use comparisons between $D(q, p)$ and $D(q, s)$. We implement these comparisons by querying the model with a query $q$ and a list of points $\{p_1, p_2, ...p_n\}$ to obtain their relative order via the model API, where the list of points is generated by our algorithm. Please refer to Algorithm 4 in Appendix for details.

In all the cases, the complexities of the high-quality model are much higher than that of the low-quality model. In the first setting, embedding a single passage takes $0.00043$ seconds when using bge-micro-v2 compared to $0.13$ seconds when using SFR-Embedding-Mistral, making the second model **>300** times slower. In the second setting, bge-micro-v2 embeddings are computed locally, while the comparisons involving the high-quality metric require calls to Gemini-2.0-Flash API, at a cost of roughly $0.01$ cents per distance evaluation, amounting to a total of $1000 to reproduce the experimental results in Figure 2.

We evaluated the retrieval quality of our approach on a benchmark collection of 6 large (i.e., of size $\geq$ one million) BEIR retrieval data sets Thakur et al. (2021). In each experiment we compared our algorithm to the standard the re-ranking approach, which retrieves the closest data items to the query with respect to $d$ and re-ranks using $D$. We observe that in almost all settings, our approach achieves a considerably better accuracy-efficiency tradeoff than re-ranking. For example, in Gemini-2.0-Flash experiments, on average, our algorithm achieves the same retrieval accuracy as re-ranking using only $\approx 200$ calls to the Gemini API, compared with $\approx 800$ calls by re-ranking, a **4x** reduction (Figure 2).

**Related Work** As described in the introduction, a popular method for utilizing a cheap metric $d$ and expensive metric $D$ in similarity search is based on "filtering" or "re-ranking". The idea is to use $d$ to construct a (long) list of candidate answers, which is then filtered using $D$ (Matveeva et al., 2006). It is a popular approach in many applications, including recommendation systems (Liu et al., 2022) and computer vision (Zhong et al., 2017). Due to the popularity of this method, we use it as a baseline in our experiments.

In addition to the re-ranking method, multiple other papers proposed different methods for combining accurate and cheap metrics to improve similarity search and related problems. We discuss those papers in more detail below. We note that, with the exception of Moseley et al. (2021); Silwal et al. (2023); Bateni et al. (2024), those methods do not appear to come with provable correctness or efficiency guarantees, or generally applicable frameworks (in contrast to the proposal in this paper). Furthermore, the three aforementioned papers (Moseley et al., 2021; Silwal et al., 2023; Bateni et al., 2024) focus on various forms of clustering, not on similarity search. The paper Moseley et al. (2021) is closest to our work, as it uses approximate nearest neighbor as a subroutine when computing the clustering. However, their algorithm only achieves the (lower) accuracy of the cheaper model, while our algorithms retains the (higher) accuracy of the expensive one.

There are also several other empirical works on similarity search that combine cheap and expensive metrics, none of which fully capture our framework to the best of our knowledge. The aforementioned paper Jayaram Subramanya et al. (2023) describes (in section 3.1) an optimization which uses the ground truth metric $D$ during the indexing phase, and proxy metric $d$, obtained via product quantization Jegou et al. (2010) during the search phase. In contrast, our framework uses $D$ during the *search phase* and $d$ during *indexing*. This difference seems crucial to our ability of providing strong approximation guarantees for the reported points. In another paper Chen et al. (2023), the authors use the proxy metric $d$ obtained by "sketching" $D$ during the query answering phase, in order to prune some points from the search queue without resorting to computing $D$. However, the data structure index is still constructed using the expensive metric $D$, as opposed the proxy metric $d$ as in our framework, which makes preprocessing more expensive in terms of space and time. Finally, Morozov & Babenko (2019) present a method for constructing a similarity graph with respect to an approximate distance function derived from a complex one; during the query phase the graph is explored using a more complex relevance function. However, their algorithm uses specific proxy

metric derived from the expensive one; in contrast, our framework allows arbitrary distance functions $d$ and $D$, as long as the distortion $C$ between them is bounded. We discuss further related work pertaining to graph-based algorithms for similarity search in Appendix A.

## 2 PRELIMINARIES

**Nearest neighbor search**   We first consider the standard formulation of *exact* nearest neighbor search. Here, we are given a set of points $P$, which is a subset of the set of all points $X$ (e.g., $X = \mathbb{R}^{dim}$). In addition, we are given access to a metric function $D$ that, for any pair of points $p, q \in X$ returns the dissimilarity between $p$ and $q$. The goal of the problem is to build an index structure that, given a *query* point $q \in X$, returns $p^* \in P$ such that $p^* = \arg\min_{p \in P} D(q, p)$. The formulation is naturally extended to more general settings, such as:

- $(1 + \varepsilon)$-approximate nearest neighbor search, where the goal is to find any $p^* \in P$ such that $D(q, p^*) \leq (1 + \varepsilon) \min_{p \in P} D(q, p)$.

- $k$-nearest neighbor search, where the goal is to find the set of $k$ nearest neighbors of $q$ in $P$ with respect to $D$. If the algorithm returns a set $S'$ of size $k$ different than the set $S$ of true $k$ nearest neighbor, the answer quality is measured via Recall or NDCG score (Järvelin & Kekäläinen, 2002).

---

**Bi-metric framework**

In our framework, we assume that we are given *two* metrics over $X$:

- The *ground truth* metric $D$, which for any pair of points $p, q \in X$ returns the "true" dissimilarity between $p$ and $q$. The metric $D$ plays the same role as in the standard nearest neighbor search problem.

- The *proxy* metric $d$, which provides a cheap approximation to the ground truth metric.

**Objective**: return nearest neighbors with respect to the expensive metric $D$; the metric $d$ is used as a proxy, in order to minimize the number of calls to the expensive metric $D$.

**Cost model**: We assume that the algorithm for constructing the data structure can use the proxy metric $d$, but *not* the ground truth metric $D$. On the other hand, the algorithm for answering a query $q$ has access to *both* metrics. However, the complexity of the query-answering procedure is measured by counting only the number of evaluations of the expensive metric $D$.

---

As described in the introduction, the above formulation is motivated by the following considerations:

- In many scenarios, evaluating the ground truth metric $D$ can be very expensive, due to factors such as model size or monetary costs associated with querying proprietary models from industry. For example, a typical call to Gemini-2.0-Flash costs roughly $0.01$ cents per distance evaluation. For SFR-Embedding-Mistral (Meng et al., 2024), it takes an A100 gpu around 196 hours to compute the embeddings of 5 million passages from the HotpotQA dataset and these embeddings occupy 83GB of disk storage; meanwhile, using the cheap model bge-micro (AI, 2023), computing these embeddings only takes 0.62 hours and 7GB of disk storage. (As a comparison, the graph index size of 5 million points occupies roughly 1GB of disk storage.) Therefore, our cost model for the query answering procedure only accounts for the number of expensive evaluations.

- In other settings, a cheap proxy metric $d$ can be obtained by approximating the ground truth metric $D$, i.e., by using product quantization (Jegou et al., 2010).

- In applications that use similarity search data structures in model training, the metric $D$ can change after each model update, necessitating re-computing embeddings and the search index over the entire database. Since this is expensive, some works (e.g., Borgeaud et al. (2022)) freeze the parts of the model that compute embeddings to avoid the computational cost of updating the data structure. Our framework offers an alternative approach, where one constructs a stable index for a proxy $d$ using frozen embeddings, but uses the up-to-date model to compute the ground-truth metric $D$ when answering nearest neighbor queries.

**Design approach:** On a high-level, the algorithms studied in this paper follow the same design pattern. Specifically, we use a graph-based nearest neighbor search algorithm, which uses calls to a metric as a black box, as a starting point. During preprocessing, the algorithm uses the proxy metric $d$. However, during the query phase, the algorithm makes calls to the accurate metric $D$. We show that, despite this "metric switch", the resulting algorithm can report provably accurate nearest neighbors with respect to the accurate metric $D$. This basic approach is then modified to achieve better performance, in theory and in practice. We apply this design approach to two popular graph-based algorithms: DiskANN and Cover Tree, but in principle any other graph-based algorithm can also be used. (We choose these two algorithms because both have provable correctness & performance guarantees, making it possible for us to obtain provable guarantees for our methods as well.)

**Assumptions about metrics:** Clearly, if the metrics $d$ and $D$ are not related to each other, the data structure constructed using $d$ alone does not help with the query retrieval. Therefore, we assume that the two metrics are related through the following definition.

**Definition 2.1.** Distance function $d$ is a $C$-approximation[2] of $D$ if for all $x, y \in X$, $d(x, y) \leq D(x, y) \leq C \cdot d(x, y)$. (1)

For a fixed metric $d$ and any point $p \in X$, radius $r > 0$, we use $B(p, r)$ to denote the ball with radius $r$ centered at $p$, i.e. $B(p, r) = \{q \in X : d(p, q) \leq r\}$. In our paper, the notion of *doubling-dimension* is central. It is a measure of intrinsic dimensionality of datasets which is popular in analyzing high dimensional datasets, especially in the context of nearest neighbor search algorithms (Gupta et al., 2003; Krauthgamer & Lee, 2004a;b; Beygelzimer et al., 2006; Indyk & Naor, 2007; Har-Peled & Kumar, 2013; Narayanan et al., 2021; Indyk & Xu, 2023). Furthermore, it is known Krauthgamer & Lee (2004b) that the complexity of approximate nearest neighbor algorithms that work for general metrics must depend the doubling dimension.

**Definition 2.2** (Doubling Dimension). $X$ has doubling dimension $\lambda_d$ with respect to metric $d$ if for any $p \in X$, and radius $r > 0$, $X \cap B(p, 2r)$ can be covered by at most $2^{\lambda_d}$ balls with radius $r$.

For a metric $d$, $\Delta_d$ is the *aspect ratio* of the input set, i.e., $\Delta_d = \text{Diam}_d / C_d$, where $\text{Diam}_d$ is the maximum distance between any pairs of points under $d$, and $C_d$ is the distance between the pair of closest distinct points under $d$. Note that both Definition 2.1 and 2.2 are only used in our theoretical analysis. Our experimental results verify that the algorithms inspired by our bi-metric framework can yield performance speedups *without any assumptions*; see Section 4.

## 3 THEORETICAL ANALYSIS

We instantiate our *bi-metric* framework for two popular nearest neighbor search algorithms: DiskANN and Cover Tree. We note that, if we treat the proxy data structure as a *black box*, we can only guarantee that it returns a $C$-approximate nearest neighbor with respect to $D$. Our theoretical analysis overcomes this, and shows that calling $D$ a sublinear number of times in the query phase (for DiskANN and Cover Tree) allows us to obtain *arbitrarily accurate* neighbors for $D$.

At a high level, the unifying theme of the algorithms that we analyze is that they both crucially use the concept of a *net*: given a parameter $r$, a $r$-net is a small subset of the dataset guaranteeing that every data point is within distance $r$ to the subset in the net. Both algorithms (implicitly or explicitly), construct nets of various scales $r$ which help route queries to their nearest neighbors in the dataset. The key insight is that a net of scale $r$ for metric $d$ is also a net under metric $D$, but with the larger scale $Cr$. Thus, if we construct smaller nets for metric $d$, they can also function as nets for the expensive metric $D$. Theoretically, this is where the advantage of our method comes from, but care must be taken to formalize the intuition.

We remark that the intuition we gave clearly does not generalize for nearest neighbor algorithms which are fundamentally different, such as locality sensitive hashing. In fact, in Appendix F *we theoretically show that such a result is impossible to achieve for LSH*. We present the analysis of DiskANN below. The analysis of Cover Tree is more complex, and hence deferred to Appendix B.

---

[2]Please see Section 4 and Figure 6 for empirical estimates of $C = D/d$. For all datasets, $C = O(1)$ for most pairs, justifying the use of this assumption.

**Preliminaries for DiskANN.** First, some helpful background is given. First we only deal with a single metric $d$. We first need the notion of an $\alpha$-shortcut reachability graph. Intuitively, it is an unweighted graph $G$ where the vertices correspond to points of a dataset $X$ such that nearby points (geometrically) are close in graph distance. The main analysis of Indyk & Xu (2023) shows that (the 'slow preprocessing version' of ) DiskANN outputs an $\alpha$-shortcut reachability graph (Theorem A.1).

**Definition 3.1** ($\alpha$-shortcut reachability Indyk & Xu (2023)). *Let $\alpha \geq 1$. We say a graph $G = (X, E)$ is $\alpha$-shortcut reachable from a vertex $p$ under a given metric $d$ if for any other vertex $q$, either $(p, q) \in E$, or there exists $p'$ s.t. $(p, p') \in E$ and $d(p', q) \cdot \alpha \leq d(p, q)$. We say a graph $G$ is $\alpha$-shortcut reachable under metric $d$ if $G$ is $\alpha$-shortcut reachable from any vertex $v \in X$.*

Given an $\alpha$-reachability graph on dataset $X$ and a query $q$, Indyk & Xu (2023) show that the greedy search procedure of Algorithm 1 (given in Appendix A) finds accurate nearest neighbor of $q$ in $X$.

**Theorem 3.2** ( Indyk & Xu (2023)). *For $\varepsilon \in (0, 1)$, there exists an $\Omega(1/\varepsilon)$-shortcut reachable graph index for a metric $d$ with max degree $Deg \leq (1/\varepsilon)^{O(\lambda_d)} \log(\Delta_d)$ (guaranteed by Theorem A.1). For any query $q$, Algorithm 1 on this graph index finds a $(1 + \varepsilon)$ nearest neighbor of $q$ in $X$ (under metric $d$) in $S \leq O(\log(\Delta_d))$ steps and makes at most $S \cdot Deg \leq (1/\varepsilon)^{O(\lambda_d)} \log(\Delta_d)^2$ calls to $d$.*

We are now ready to state the main theorem of this section.

**Theorem 3.3.** *Let $Q_{\texttt{DiskAnn}}(\varepsilon, \Delta_d, \lambda_d) = (1/\varepsilon)^{O(\lambda_d)} \log(\Delta_d)^2$ denote the query complexity of the DiskANN data structure[3], where we build and search using the same metric $d$. Consider two metrics $d$ and $D$ satisfying Equation 1. Suppose we build an $C/\varepsilon$-shortcut reachability graph $G$ using Theorem A.1 for metric $d$, but search using metric $D$ in Algorithm 1 for a query $q$ with $L = 1$. Then:*

1. *The space used by $G$ is at most $n \cdot (C/\varepsilon)^{O(\lambda_d)} \log(\Delta_d)$.*
2. *Running Algorithm 1 using $D$ finds a $1 + \varepsilon$ nearest neighbor of $q$ in the dataset $X$ (under $D$).*
3. *On any query $q$, Algorithm 1 invokes $D$ at most $Q_{\texttt{DiskAnn}}(\varepsilon/C, C\Delta_d, \lambda_d)$.*

To prove the theorem, we first show that a shortcut reachability graph of $d$ is also a shortcut reachability graph of $D$, albeit with slightly different parameters, with a proof in Appendix C.

**Lemma 3.4.** *Suppose metrics $d$ and $D$ satisfy relation (1). Suppose $G = (X, E)$ is $\alpha$-shortcut reachable under $d$ for $\alpha > C$. Then $G = (X, E)$ is an $\alpha/C$-shortcut reachable under $D$.*

*Proof of Theorem 3.3.* By Lemma 3.4, the graph $G = (X, E)$ constructed for metric $d$ is also a $O(1/\varepsilon)$ reachable for the other metric $D$. Then we simply invoke Theorem 3.2 for a $(1/\varepsilon)$-reachable graph index for metric $D$ with degree limit $Deg \leq (C/\varepsilon)^{O(\lambda_d)} \log(\Delta_d)$ and the number of greedy search steps $S \leq O(\log(C\Delta_d))$. Thus the total number of $D$ distance call bounded by $(C/\varepsilon)^{O(\lambda_d)} \log(C\Delta_d)^2 \leq Q_{\texttt{DiskAnn}}(\varepsilon/C, C\Delta_d, \lambda_d)$. This proves the accuracy bound as well as the number of calls we make to metric $D$ during the greedy search procedure of Algorithm 1. The space bound follows from Theorem A.1, since $G$ is a $C/\varepsilon$-reachability graph for metric $d$. □

# 4 EXPERIMENTS

The starting point of our implementation is the DiskANN based algorithm from Theorem 3.3, which we engineer to optimize performance[4]. We compare it to two other methods on all large BEIR retrieval tasks (Thakur et al., 2021), i.e., for datasets with corpus size $> 10^6$, see below.

**Methods** We evaluate the following methods. $\mathcal{Q}$ denotes the query budget, i.e., the maximum number of calls an algorithm can make to $D$ during a query. We vary $\mathcal{Q}$ in our experiments.

• Bi-metric (our method): We build a graph index with the cheap distance function $d$ (we discuss our choice of graph indices in the experiments shortly). Given a query $q$, we first search for $q$'s top-$\mathcal{Q}/2$ nearest neighbor under metric $d$. Then, we start a second-stage search from the $\mathcal{Q}/2$ returned vertices using distance function $D$ on the same graph index until we reach the quota $\mathcal{Q}$. We report the 10 closest neighbors seen so far by distance function $D$.

---

[3]I.e., the upper bound on the number of calls made to $d$ on any query

[4]Our experiments are run on 56 AMD EPYC-Rome processors with 400GB of memory and 4 NVIDIA RTX 6000 GPUs. Our experiment in Figure 1 takes roughly 3 days.

• Bi-metric (baseline): This is the standard retrieve-then-rerank method that is widely popular. We build a graph index with the cheap distance function $d$. Given a query $q$, we first search for $q$'s top-$\mathcal{Q}$ nearest neighbor under metric $d$. As explained below, we can assume that empirically the first step returns the *true* top-$\mathcal{Q}$ nearest neighbors under $d$. Then, we calculate distance using $D$ for all the $\mathcal{Q}$ returned vertices and report the top-10.

• Single metric: This is the standard nearest neighbor search with a single distance function $D$. We build the graph index directly with the expensive distance function $D$. Given a query $q$, we do a standard greedy search to get the top-10 closest vertices to $q$ with respect to distance $D$ until we reach quota $\mathcal{Q}$. We ignore the large number of $D$ distance calls in the indexing phase and only count the quota in the search phase. Note that this method doesn't satisfy our "bi-metric" formulation as it uses an extensive number of $D$ distance calls ($\Omega(n)$ calls) in index construction. However, we implement it for comparison since it represents a natural baseline, if one does not care about the prohibitively large number of calls made to $D$ during index building.

For both Bi-metric methods (ours and baseline), in the first-stage search under distance $d$, we initialize the parameters of the graph index so that empirically, it returns the true nearest neighbors under distance $d$. This is done by setting the 'queue length' parameter $L$ to be 30000 for datasets with corpus size $> 10^6$ (Climate-FEVER (Diggelmann et al., 2020), FEVER (Thorne et al., 2018), HotpotQA (Yang et al., 2018), MSMARCO (Bajaj et al., 2018), NQ (Kwiatkowski et al., 2019), DBPedia (Hasibi et al., 2017)). Our choice of $L$ is large enough to ensure that the returned vertices are almost true nearest neighbors under distance $d$. For example, the standard parameters used are a factor of 10 smaller. We also empirically verified that the nearest neighbors returned for $d$ with such large values of $L$ corroborated with published BEIR benchmark values [5].

**Datasets** We experiment with all 6 BEIR retrieval datasets of size $>10^6$ (Climate-FEVER (Diggelmann et al., 2020), FEVER (Thorne et al., 2018), HotpotQA (Yang et al., 2018), MSMARCO (Bajaj et al., 2018), NQ (Kwiatkowski et al., 2019), DBPedia (Hasibi et al., 2017)). We report the results on these dataests' test split, except for MSMARCO where we report the results on its dev split.

**Embedding Models** We select a highly ranked model "SFR-Embedding-Mistral" as our expensive model to provide groundtruth metric $D$. Meanwhile, we select three models on the pareto curve of the BEIR retrieval size-average score plot to test how our method performs under different model scale combinations. These three small models are "bge-micro-v2", "gte-small", "bge-base-en-v1.5". Please refer to Table 1 for details. As described earlier, both metrics $d(p, q)$ and $D(p, q)$ are induced by the Euclidean distance between the embeddings of $p$ and $q$ using the respective models. The embeddings defining the proxy metric $d$ are pre-computed and stored during the pre-processing, and then used to construct the data structure. The embeddings defining the accurate metric $D$ are computed on the fly during the query processing stage. Specifically, to answer a query $q$, the algorithm first computes the embedding $f(q)$ of $q$. Then, whenever the value of $D(q, p)$ is needed, the algorithm computes $f(p)$ and evaluates $D(p, q) = \|f(q) - f(p)\|$. Thus, the cost of evaluating $D(p, q)$ is equal to the cost of embedding $p$. (In other scenarios where $D(p, q)$ is evaluated using a proprietary system over an API call, the cost is determined by the vendor's prices and/or the network speed.).

| Model Name | Embedding Dimension | Model Size | BEIR Retrieval Score |
|---|---|---|---|
| SFR-Embedding-Mistral (Meng et al., 2024) | 4096 | 7111M | 59 |
| bge-base-en-v1.5 (Xiao et al., 2023) | 768 | 109M | 53.25 |
| gte-small (Li et al., 2023) | 384 | 33M | 49.46 |
| bge-micro-v2 (AI, 2023) | 384 | 17M | 42.56 |

Table 1: Different models used in our experiments

**Nearest Neighbor Search Algorithms** The search algorithms we employ in our experiments are DiskANN (Jayaram Subramanya et al., 2019) and NSG (Fu et al., 2019a). We use standard parameter choices for both; see Appendix E.

**Metric** Given a fixed expensive distance function quota $\mathcal{Q}$, we report the accuracy of retrieved results for different methods. We always insure that all algorithms never use more than $\mathcal{Q}$ expensive distance computations. Following the BEIR retrieval benchmark, we report the NDCG@10 score. In addition, we also report Recall@10, where the ground truth is defined by the 10 nearest neighbors *according*

---

[5]from `https://huggingface.co/spaces/mteb/leaderboard`

*to the expensive metric $D$; this is akin to the methodology used to evaluate approximate nearest neighbor search algorithms.*

## 4.1 EXPERIMENT RESULTS AND ANALYSIS

Please refer to Figure 1 for our results with $d$ distance function set to "bge-micro-v2" and $D$ set to "SFR-Embedding-Mistral", with the underlying graph index being DiskANN. To better focus on the convergence speed of different methods toward the "Single metric (limit)" (perfect nearest neighbor retrieval with respect to $D$), we cut off the y-axis at a relatively high accuracy, so some curves may not start from x equals 0 if their accuracy doesn't reach the threshold. We observe that our method converges to the optimal accuracy much faster than bi-metric (baseline) and single metric in most cases. For example for HotpotQA, the NDCG@10 score achieved by the baselines for 8000 calls to $D$ is comparable to our method, using less than 2000 calls to $D$, leading to $>$ **4x** fewer evaluations of the expensive model. This leads to substantial time savings. For example, consider our largest data set HotpotQA. The first stage of the query answering procedure (using $d$) takes only 0.37s per query $q$, while each evaluation of $D(p,q)$ during the second stage takes 0.13s; this translates into roughly 260s per query when 2000 evaluations of $D$ are used. In contrast, the baseline method requires 8000 calls to $D$, which translates into a cost of roughly 1040s per query.

This means that utilizing the graph index built for the distance function proxy to perform a greedy search using $D$ is more efficient than naively iterating the returned vertex list to re-rank using $D$ (baseline). Also note that our method converges faster than "Single metric" in all the datasets. This phenomenon happens even if "Single metric" is allowed infinite expensive distance function calls in its indexing phase to build the ground truth graph index. This suggests that the quality of the underlying graph index is not as important, and the early routing steps in the searching algorithm can be guided with a cheap distance proxy functions to save expensive distance function calls.

Similar conclusion holds for the recall plot (see Figure 7), where our method has an even larger advantage over Bi-metric (baseline) and is better than the Single metric in most cases, except FEVER and HotpotQA. We report the results of using different model pairs, using the NSG algorithm as our graph index, and measuring Recall@10 in Appendix E. Please see ablation studies in Appendix D.

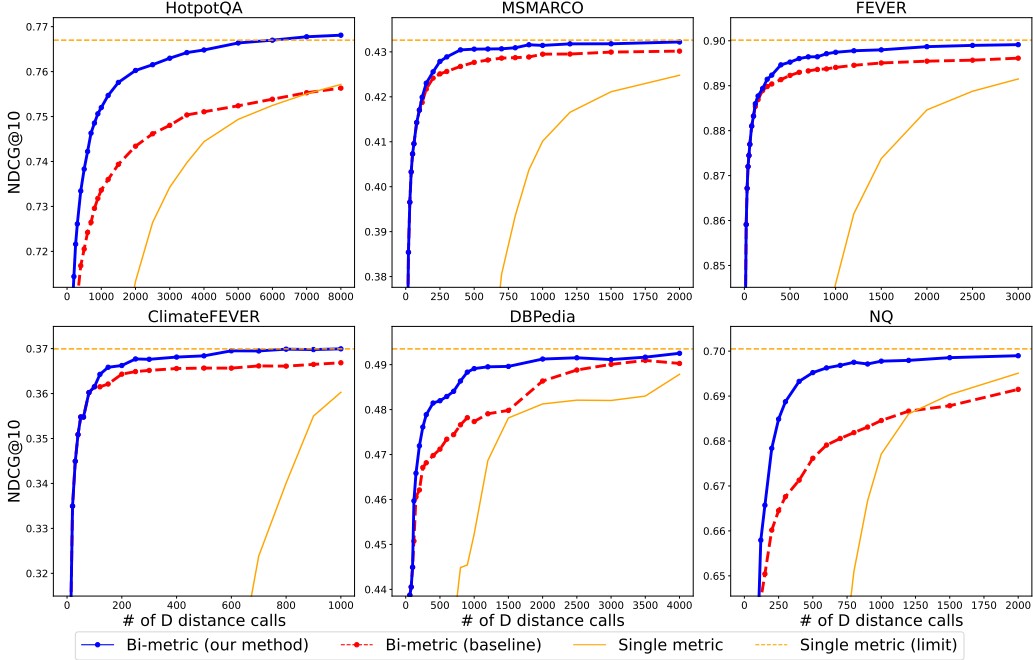

Figure 1: Results for 6 BEIR Retrieval datasets. The x-axis is the number of expensive distance function calls. The y-axis is the NDCG@10 score. The cheap model is "bge-micro-v2", the expensive model is "SFR-Embedding-Mistral", and the nearest neighbor search algorithm used is DiskANN.

Lastly, we measure the empirical value of $C$ (the relationship between $d/D$ from (1)). For simplicity, we assumed that $d \leq D \leq C \cdot d$ for $C \geq 1$ in (1) in our theoretical bounds. This is without loss

of generality by scaling. In practice, we observe that the ratio of distances $C := D/d$ is always clustered around one. For example, if we use "SFR-Embedding-Mistral" to provide the distance $D$, and "bge-micro-v2" to provide the distance $d$, then for HotpotQA, we empirically found that $99.9\%$ of $10^5$ randomly sampled pairs satisfy $0.6 \leq C \leq 1.5$. We observed the same qualitative behavior for our other datasets; see Figure 6 in Appendix E.

## 4.2 APPLICATION TO A LLM-BASED LISTWISE RERANKER

Following the method proposed by Sun et al. (2023), recently, there has been a trend to use LLMs to re-rank passages. Though the score output by a re-ranker does not meet the definition of a metric, our algorithm still works in this scenario. We prompt *Gemini-2.0-Flash* to re-rank different passages based on their relevance to a search query. We slightly modify the search algorithm (Algorithm 4 in Appendix), as now the re-ranker only returns an order rather than independent relevance scores. Since querying proprietary models like Gemini-2.0-Flash is expensive, we only use 500 queries randomly sampled from the query sets. The averaged results for all 6 data sets are in Figure 2. The results on individual dataset and other experimental details are in Appendix E. We can observe that our bi-metric framework yields good results in this setting. Our method achieves higher NDCG@10 scores while sending fewer passages to the re-ranker. (The slight perturbation near the end of the curves is because of the LLM's occasional mistakes in judging the order of different passages.)

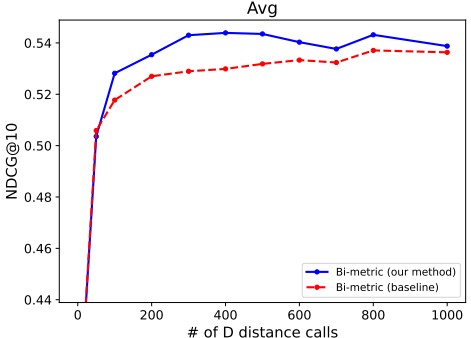

Figure 2: Average results for 6 BEIR Retrieval datasets. The x-axis is the number of passages sent to the reranker. The y-axis is the NDCG@10 score. The cheap distance function is provided by "bge-micro-v2", the expensive distance comparator is "Gemini-2.0-Flash", and the nearest neighbor search algorithm used is DiskANN.

## 5 CONCLUSION

We presented a new framework for designing nearest neighbor algorithms that use two metrics: a *ground truth* metric $D$ that defines the true nearest neighbors, and a *proxy* metric $d$ which provides a cheap but imperfect approximation to the ground truth. Our theoretical results show that, as long as $d$ approximates $D$ up to some constant $C > 1$, a nearest neighbor data structure constructed using the proxy metric $d$ can return nearest neighbors with respect to $D$ up to arbitrarily small approximation $1 + \varepsilon$ in sub-linear time, as long as the ground truth metric $D$ is used during the query answering phase. This improves over an approximation of $C$ offered by the standard re-ranking approach, which retrieves $k$ nearest neighbors with respect to $d$, and then scans them to retrieve the true nearest neighbor with respect to $D$. Experimentally, we show that our method offers an improved accuracy-efficiency tradeoff in settings where $d$ and $D$ are computed using embeddings or LLMs of vastly different complexities, for BEIR text retrieval benchmarks.

Our framework requires that the (cheap) proxy metric $d$ provides a "reasonable" approximation to the (expensive) ground-truth metric $D$. The practical effectiveness of our approach, as validated by our empirical results on the BEIR benchmark, suggests that such related metrics are readily available for real-world datasets. However, the framework's performance may degrade if the proxy metric is a poor approximation of the ground truth. We note that this is an inherent limitation of working with a proxy metric, since in the extreme case $d$ might provide no useful information about $D$. However, our theorems always guarantees a $1 + \varepsilon$ approximate solution for any constant $C$, with query time depending on $C$.

Our results hold only for graph-based nearest neighbor data structures, and not for other algorithms such as LSH. We show that this is a fundamental limitation of LSH itself: in Appendix F, we theoretically show that LSH-type algorithms cannot take advantage of a proxy metric.

Finally, we recognize that adapting the framework to new domains may require some implementation-specific adjustments. For example, when applying our method to an LLM-based listwise reranker, we had to modify the search algorithm to accommodate a relative ordering instead of a strict distance score. We believe this is a testament to flexibility offered by our framework.

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

## A  FURTHER RELATED WORKS

**Graph-based algorithms for similarity search**  The algorithms studied in this paper rely on graph-based data structures for (approximate) nearest neighbor search. Such data structures work for general metrics, which, during the pre-processing, are approximated by carefully constructed graphs. Given the graph and the query point, the query answering procedure greedily searches the graph to identify the nearest neighbors. Graph-based algorithms have been extensively studied both in theory Krauthgamer & Lee (2004a); Beygelzimer et al. (2006) and in practice Fu et al. (2019b); Jayaram Subramanya et al. (2019); Malkov & Yashunin (2018); Harwood & Drummond (2016). See Clarkson et al. (2006); Wang et al. (2021) for an overview of these lines of research.

**Theorem A.1** (Indyk & Xu (2023)). *Given a dataset $X$, $\alpha \geq 1$, and fixed metric $d$ the slow preprocessing DiskANN algorithm (Algorithm 4 in Indyk & Xu (2023)) outputs a $\alpha$-shortcut reachability graph $G$ on $X$ as defined in Definition 3.1 (under metric $d$). The space complexity of $G$ is $n \cdot \alpha^{O(\lambda_d)} \log(\Delta_d)$.*

---

**Algorithm 1** GreedySearch($q, d, L$)

---

1: **Input**: Graph index $G = (X, E)$, distance function $d$, starting point $s$, query point $q$, queue length limit $L$
2: **Output**: visited vertex list $U$
3: $A \leftarrow \{s\}$
4: $U \leftarrow \varnothing$
5: **while** $A \setminus U \neq \varnothing$ **do**
6:     $v \leftarrow \operatorname{argmin}_{v \in A \setminus U} d(x_v, q)$
7:     $A \leftarrow A \cup Neighbors(v)$                          ▷ Neighbors in $G$
8:     $U \leftarrow U \cup v$
9:     **if** $|A| > L$ **then**
10:         $A \leftarrow$ top L closest vertex to $q$ in $A$
11: sort $U$ in increasing distance from $q$
12: **return** $U$

---

## B  ANALYSIS OF COVER TREE

We now analyze Cover Tree under the bi-metric framework. First, some helpful background is presented below.

### B.0.1  PRELIMINARIES FOR COVER TREE

The notion of a cover is central. We specialize it to the greedy cover used in the Cover Tree datastructure.

**Definition B.1** (Greedy Cover Construction). A $r$-cover $\mathcal{C}$ of a set $X$ given a metric $d$ is defined as follows. Initially $\mathcal{C} = \emptyset$. Run the following two steps until $X$ is empty.

1. Pick an arbitrary point $x \in X$ and remove $B(x, r) \cap X$ from $X$.

2. Add $x$ to $\mathcal{C}$.

Note that a cover with radius $r$ satisfies the following two properties: every point in $X$ is within distance $r$ to some point in $\mathcal{C}$ (under the same metric $d'$), and all points in $\mathcal{C}$ are at least distance $r$ apart from each other.

We now introduce the cover tree datastructure of Beygelzimer et al. (2006). For the data structure, we create a sequence of covers $\mathcal{C}_{-1}, \mathcal{C}_0, \ldots$. Every $\mathcal{C}_i$ is a layer in the final Cover Tree $\mathcal{T}$.

---

**Algorithm 2** Cover Tree Data structure

---

1: **Input:** A set $X$ of $n$ points, metric $d$, real number $T \geq 1$.
2: **Output:** A tree on $X$
3: **procedure** COVER-TREE$(d, T)$
4:    WLOG, all distances between points in $X$ under $d$ are in $(1, \Delta]$ by scaling.
5:    $\mathcal{C}_{-1} = \mathcal{C}_0 = X$
6:    Define $\mathcal{C}_i$ as a $2^i/T$-cover of $\mathcal{C}_{i-1}$ for any $i > 0$ under metric $d$
7:    $\mathcal{C}_i \subseteq \mathcal{C}_{i-1}$ for all $i > 0$.
8:    $t = O(\log(\Delta T))$                    ▷ $t$ is the number of levels of $\mathcal{T}$
9:    **for** $i = -1$ to $t$ **do**
10:        $\mathcal{C}_i$ corresponds to tree nodes of $\mathcal{T}$ on level $i$
11:        Each $p \in \mathcal{C}_{i-1} \setminus \mathcal{C}_i$ is connected to exactly one $p \in \mathcal{C}_i$ such that $d(p, p') \leq 2^i/T$
12:    **Return** tree $\mathcal{T}$

---

The following result about the space bound of the datastructure is from Beygelzimer et al. (2006) and we to Beygelzimer et al. (2006) for more details about the space bound.

**Lemma B.2** (Theorem 1 in Beygelzimer et al. (2006)). *$\mathcal{T}$ takes $O(n)$ space, regardless of the value of $r$.*

*Proof.* We use the *explicit* representation of $\mathcal{T}$ (as done in Beygelzimer et al. (2006)), where we coalesce all nodes in which the only child is a self-child. The underlying idea is simple: the covers are nested (a smaller scale cover contains all larger scale covers). Thus, a node in the tree has children that also correspond to the same net point. The explicit representation of the tree simply collapses all long paths in the tree (since these correspond to the same net point). Thus, every node in this compressed tree has a parent that represents a different net point and a child that represents a different net point. This can be used to show that there are $O(n)$ edges in total in the tree, independent of all other parameters. □

We note that it is possible to construct the cover tree data structure of Algorithm 2 in time $2^{O(\lambda_d)} n \log n$, but it is not important to our discussion Beygelzimer et al. (2006).

Now we describe the query procedure. Here, we can query with a metric $D$ that is possibly different than the metric $d$ used to create $\mathcal{T}$ in Algorithm 2.

---

**Algorithm 3** Cover Tree Search

---

1: **Input:** Cover tree $\mathcal{T}$ associated with point set $X$, query point $q$, metric $D$, accuracy $\varepsilon \in (0, 1)$.
2: **Output:** A point $p \in X$
3: **procedure** COVER-TREE-SEARCH
4:    $t \leftarrow$ number of levels of $\mathcal{T}$
5:    $Q_t \leftarrow \mathcal{C}_t$                    ▷ We use the covers that define $\mathcal{T}$
6:    $i \leftarrow t$
7:    **while** $i \neq -1$ **do**
8:        $Q = \{p \in \mathcal{C}_{i-1} : p \text{ has a parent in } Q_i\}$
9:        $Q_{i-1} = \{p \in Q : D(q, p) \leq D(q, Q) + 2^i\}$
10:        **if** $D(q, Q_{i-1}) \geq 2^i(1 + 1/\varepsilon)$ **then**
11:            Exit the while loop.
12:        $i \leftarrow i - 1$
13:    **Return** point $p \in Q_{i-1}$ that is closest to $q$ under $D$

---

### B.0.2 THE MAIN THEOREM

We construct a cover tree $\mathcal{T}$ using metric $d$ and $T$ from Equation 1 in Algorithm 2. Upon a query $q$, we search for an approximate nearest neighbor in $\mathcal{T}$ in Algorithm 3, using metric $D$ instead. Our main theorem is the following.

**Theorem B.3.** *Let $Q_{\text{CoverTree}}(\varepsilon, \Delta_d, \lambda_d) = 2^{O(\lambda_d)} \log(\Delta_d) + (1/\varepsilon)^{O(\lambda_d)}$ denote the query complexity of the standard cover tree datastructure, where we set $T = 1$ in Algorithm 2 and build and search using the same metric $d$. Now consider two metrics $d$ and $D$ satisfying Equation 1. Suppose we build a cover tree $\mathcal{T}$ with metric $d$ by setting $T = C$ in Algorithm 2, but search using metric $D$ in Algorithm 3. Then the following holds:*

1. *The space used by $\mathcal{T}$ is $O(n)$.*

2. *Running Algorithm 3 using $D$ finds a $1 + \varepsilon$ approximate nearest neighbor of $q$ in the dataset $X$ (under metric $D$).*

3. *On any query, Algorithm 3 invokes $D$ at most*

$$C^{O(\lambda_d)} \log(\Delta_d) + (C/\varepsilon)^{O(\lambda_d)} = \tilde{O}(Q_{\text{CoverTree}}(\Omega(\varepsilon/C), \Delta_d, \lambda_d)).$$

   *times.*

Two prove Theorem B.3, we need to: (a) argue correctness and (b) bound the number of times Algorithm 3 calls its input metric $D$. While both follow from similar analysis as in Beygelzimer et al. (2006), it is not in a black-box manner since the metric we used to search $\mathcal{T}$ in Algorithm 3 is different than the metric used to build $\mathcal{T}$ in Algorithm 2.

We begin with a helpful lemma.

**Lemma B.4.** *For any $p \in \mathcal{C}_{i-1}$, the distance between $p$ and any of its descendants in $\mathcal{T}$ is bounded by $2^i$ under $D$.*

*Proof.* The proof of the lemma follows from Theorem 2 in Beygelzimer et al. (2006). There, it is shown that for any $p \in \mathcal{C}_{i-1}$ the distance between $p$ and any descendant $p'$ is bounded by $d(p, p') \leq \sum_{j=-\infty}^{i-1} 2^j/T = 2^i/T$, implying the lemma after we scale by $C$ due to Equation 1 (note we set $T = C$ in the construction of $\mathcal{T}$ in Theorem B.3). $\square$

We now argue accuracy.

**Theorem B.5.** *Algorithm 3 returns a $1 + \varepsilon$-approximate nearest neighbor to query $q$ under $D$.*

*Proof.* Let $p^*$ be the true nearest neighbor of query $q$. Consider the leaf to root path starting from $p^*$. We first claim that if $Q_i$ contains an ancestor of $p^*$, then $Q_{i-1}$ also contains an ancestor $q_{i-1}$ of $p^*$. To show this, note that $D(p^*, q_{i-1}) \leq 2^i$ by Lemma B.4, so we always have

$$D(q, q_{i-1}) \leq D(q, p^*) + D(p^*, q_{i-1}) \leq D(q, Q) + 2^i,$$

meaning $q_{i-1}$ is included in $Q_{i-1}$.

When we terminate, either we end on a single node, in which case we return $p^*$ exactly (from the above argument), or when $D(q, Q_{i-1}) \geq 2^i(1 + 1/\varepsilon)$. In this latter case, we additionally know that

$$D(q, Q_{i-1}) \leq D(q, p^*) + D(p^*, Q_{i-1}) \leq D(q, p^*) + 2^i$$

since an ancestor of $p^*$ is contained in $Q_{i-1}$ (namely $q_{i-1}$ from above). But the exit condition implies

$$2^i(1 + 1/\varepsilon) \leq D(q, p^*) + 2^i \implies 2^i \leq \varepsilon D(q, p^*),$$

which means

$$D(q, Q_{i-1}) \leq D(q, p^*) + 2^i \leq D(q, p^*) + \varepsilon D(q, p^*) = (1 + \varepsilon)D(q, p^*),$$

as desired. $\square$

Finally, we bound the query complexity. The following follows from the arguments in Beygelzimer et al. (2006).

**Theorem B.6.** *The number of calls to $D$ in Algorithm 3 is bounded by $C^{O(\lambda_d)} \cdot \log(\Delta_d C) + (C/\varepsilon)^{O(\lambda_d)}$.*

*Proof Sketch.* The bound follows from Beygelzimer et al. (2006) but we briefly outline it here. The query complexity is dominated by the size of the sets $Q_{i-1}$ in Line 9 as the algorithm proceeds. We give two ways to bound $Q_{i-1}$. Before that, note that the points $p$ that make up $Q_{i-1}$ are in a cover (under $d$) by the construction of $\mathcal{T}$, so they are all separated by distance at least $\Omega(2^i/C)$ (under $d$). Let $p^*$ be the closest point to $q$ in $X$.

- **Bound 1**: In the iterations where $D(q, p^*) \leq O(2^i)$, we have the diameter of $Q_{i-1}$ under $D$ is at most $O(2^i)$ as well. This is because an ancestor $q_{i-1} \in C_{i-1}$ of $p^*$ is in $Q$ of line 8 (see proof of Theorem B.5), meaning $D(q, Q) \leq O(2^i)$ due to Lemma B.4. Thus, any point $p \in Q_{i-1}$ satisfies $D(q, p) \leq D(q, Q) + 2^i = O(2^i)$. From Equation 1, it follows that the diameter of $Q_{i-1}$ under $d$ is also at most $O(2^i)$. We know the points in $Q_{i-1}$ are separated by mutual distance at least $\Omega(2^i/C)$ under $d$, implying that $|Q_{i-1}| \leq C^{O(\lambda_d)}$ in this case by a standard packing argument. This case can occur at most $O(\log(\Delta C))$ times, since that is the number of different levels of $\mathcal{T}$.

- **Bound 2**: Now consider the case where $D(q, p^*) \geq \Omega(2^i)$. In this case, we have that the points in $Q_{i-1}$ have diameter at most $O(2^i/\varepsilon)$ from $q$ (under $D$), due to the condition of line 10. Thus, the diameter is also bounded by $O(2^i/\varepsilon)$ under $d$. By a standard packing argument, this means that $|Q_{i-1}| \leq (C/\varepsilon)^{O(\lambda_d)}$, since again $Q_{i-1}$ are mutually separated by distance at least $\Omega(2^i/C)$ under $d$. However, our goal is to show that the number of iterations where this bound is relevant is at most $O(\log(1/\varepsilon))$. Indeed, we have $D(q, Q_{i-1}) \leq O(2^i/\varepsilon)$, meaning $2^i \geq \Omega(\varepsilon D(q, Q_{i-1})) \geq \Omega(\varepsilon D(q, p^*))$ Since we are decrementing the index $i$ and are in the case where $D(q, p^*) \geq \Omega(2^i)$, this can only happen for $O(\log(1/\varepsilon))$ different $i$'s.

Combining the two bounds proves the theorem. □

The proof of Theorem B.3 follows from combining Lemmas B.2 and Theorems B.5 and B.6.

## C  OMITTED PROOFS FROM THE MAIN BODY

We give the proof of Lemma 3.4

*Proof.* Let $(p, q)$ be a pair of distinct vertices such that $(p, q) \notin E$. Then we know that there exists a $(p, p') \in E$ such that $d(p', q) \cdot \alpha \leq d(p, q)$. From relation (1), we have $\frac{1}{C} \cdot D(p', q) \cdot \alpha \leq d(p', q) \cdot \alpha \leq d(p, q) \leq D(p, q)$, as desired. □

## D  ABLATION STUDIES

We investigate the impact of different components of our experiments in Section 4. All ablation studies are run on HotpotQA dataset as it is one of the largest and most difficult retrieval dataset where the performance gaps between different methods are substantial.

**Different model pairs**  Fixing the expensive model as "SFR-Embedding-mistral" (Meng et al., 2024), we experiment with 2 other cheap models from the BEIR retrieval benchmark: "gte-small" Li et al. (2023) and "bge-base" Xiao et al. (2023). These models have different sizes/capabilities, summarized in Table 1. For complete results on all 6 BEIR Retrieval datasets for different cheap models, we refer to Figures 8, 9, 10, and 11 in Appendix E. Here, we only focus on HotpotQA.

From Figure 3, we can observe that the improvement of our method is most substantial when there is a large gap between the qualities of the cheap and expensive models. This is not surprising: If the cheap model has already provided enough accurate distances, simple re-ranking can easily get to the optimal retrieval results with only a few expensive distance calls. Note that even in the latter case, our second-stage search method still performs at least as good as re-ranking. Therefore, we believe that the ideal scenario for our method is a small and efficient model deployed locally, paired with a remote large model accessed online through API calls to maximize the advantages of our method.

**Varying neighbor search algorithms**   We implement our method with another popular empirical nearest neighbor search algorithm called NSG (Fu et al., 2019b). We obtain the same qualitative behavior as DiskANN, with details given in Section E.

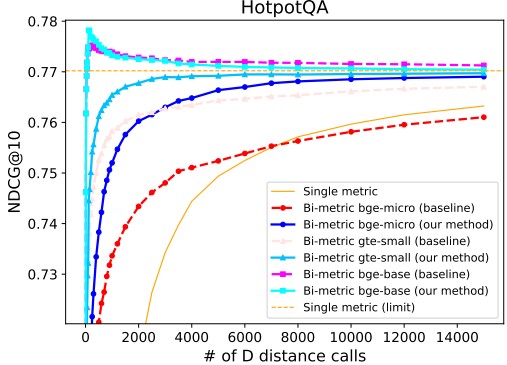
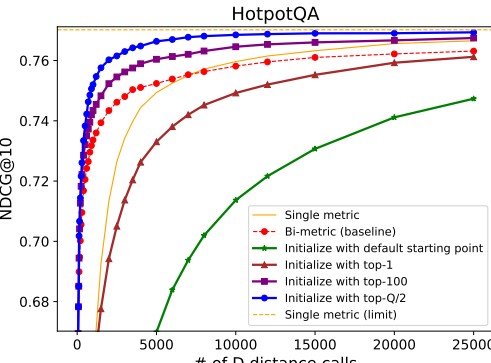

Figure 3: HotpotQA test results for different models as the distance proxy. Blue / skyblue / cyan curves represent Bi-metric (our method) with bge-micro / gte-small / bge-base models. Red / rose / magenta curves represent Bi-metric (baseline) with bge-micro / gte-small / bge-base models

Figure 4: HotpotQA test results for different search initializations for the second-stage search of Bi-metric (our method). Blue / purple / brown / green curves represent initializing our second-stage search with top-$\mathcal{Q}/2$, top-100, top-1, or the default vertex.

**Impact of the number of starting points**   In the second-stage search of our method, we start from multiple points returned by the first-stage search via the cheap distance metric. We investigate how varying the starting points for the second-stage search impact the final results. We try four different setups:

- Default: We start a standard nearest neighbor search using metric $D$ from the default entry point of the graph index, which means that we don't use the first stage search.

- Top-$K$ points retrieved by the first stage search: Suppose our expensive distance calls quota is $\mathcal{Q}$. We start our second search from the top $K$ points retrieved by the first stage search. We experiment with the following different choices of $K$: $K_1 = 1$, $K_{100} = 100$, $K_{\mathcal{Q}/2} = \max(100, \mathcal{Q}/2)$ (note $K_{\mathcal{Q}/2}$ is the choice we use in Figure 1).

From Figure 4, we observe that utilizing results from the first-stage search helps the second-stage search to find the nearest neighbor quicker. For comparison, we experiment with initializing the second-stage search from the default starting point (green), which means that we don't need the first-stage search and only use the graph index built from $d$ (cheap distance function). The DiskANN algorithm still manages to improve as the allowed number of $D$ distance calls increases, but it converges the slowest compared to all the other methods.

Using multiple starting points further speeds up the second stage search. If we only start with the top-1 point from the first stage search (brown), its NDCG curve is still worse than Bi-metric (baseline, red) and Single metric (orange). As we switch to top-100 (purple) or top-$\mathcal{Q}/2$ (blue) starting points, the NDCG curves increase evidently.

We provide two intuitive explanations for these phenomena. First, the approximation error of the cheap distance function doesn't matter that much in the earlier stage of the search, so the first stage search with the cheap distance function can quickly get to the true 'local' neighborhood without any expensive distance calls, thus saving resources for the second stage search. Second, the ranking provided by the cheap distance function is not accurate because of its approximation error, so starting from multiple points should give better results than solely starting from the top few, which also justifies the advantage of our second-stage search over re-ranking.

**Impact of the first-stage search queue length**   In our experiments, we use a very large queue length $L$ for the first-stage search to ensure that the starting point is the closest embedding with

respect to the cheap distance. Here, we perform an ablation study to evaluate how the retrieval quality changes when using smaller queue lengths. On HotpotQA and NQ, we run experiments with varying queue lengths for the first-stage search; see Figure 5. We observe that different values of $L$ impact the final retrieval quality noticeably on HotpotQA but not much on NQ. Overall, the NDCG curves become stable after $L = 1000$ for both datasets.

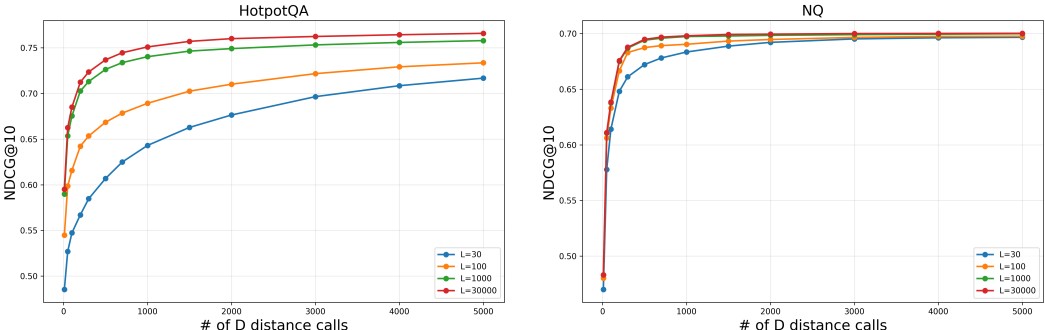

Figure 5: HotpotQA and NQ test results for different first stage queue length $L = \{30, 100, 1000, 30000\}$

# E  COMPLETE EXPERIMENTAL RESULTS

**Parameter choices for Nearest Neighbor Search algorithms**    The parameter choices for DiskANN are $\alpha = 1.2$, $l\_build = 125$, $max\_outdegree = 64$ (the standard choices used in ANN benchmarks Aumüller et al. (2020)). The parameter choices for NSG are the same as the authors' choices for GIST1M dataset (Jégou et al., 2011): $K = 400$, $L = 400$, $iter = 12$, $S = 15$, $R = 100$. NSG also requires building a knn-graph with efanna, where we use the standard parameters: $L = 60$, $R = 70$, $C = 500$.

**Empirical Results**    We report the empirical results of using different embedding models as distance proxy, using the NSG algorithm, and measuring Recall@10.

1. We report the results of using "bge-micro-v2" as the distance proxy $d$ and using DiskANN for building the graph index. See Figure 7 for Recall@10 metric plots.

2. We report the results of using "gte-small" as the distance proxy $d$ and using DiskANN for building the graph index. See Figure 8 for NDCG@10 metric plots and Figure 9 for Recall@10 metric plots.

3. We report the results of using "bge-base-en-v1,5" as the distance proxy $d$ and using DiskANN for building the graph index. See Figure 10 for NDCG@10 metric plots and Figure 11 for Recall@10 metric plots.

4. We report the results of using "bge-micro-v2" as the distance proxy $d$ and using NSG for building the graph index. See Figures 12 for NDCG@10 metric plots and 13 for Recall@10 metric plots.

We can see that for all the different cheap distance proxies ("bge-micro-v2" Xiao et al. (2023), "gte-small" Li et al. (2023), "bge-base-en-v1.5" Xiao et al. (2023)) and both nearest neighbor search algorithms (DiskANN Jayaram Subramanya et al. (2019) and NSG Fu et al. (2019b)), our method has better NDCG and Recall results on most datasets. Moreover, naturally the advantage of our method over Bi-metric (baseline) is larger when there is a large gap between the qualities of the cheap distance proxy $d$ and the ground truth distance metric $D$. This makes sense because as their qualities converge, the cheap proxy alone is enough to retrieve the closest points to a query for the expensive metric $D$.

We also report the histograms of empirical $C = d/D$ values using "bge-micro-v2' as the distance proxy $d$ in Figure 6. For all 6 datasets, the distance ratio $C = d/D$ concentrates well around 1

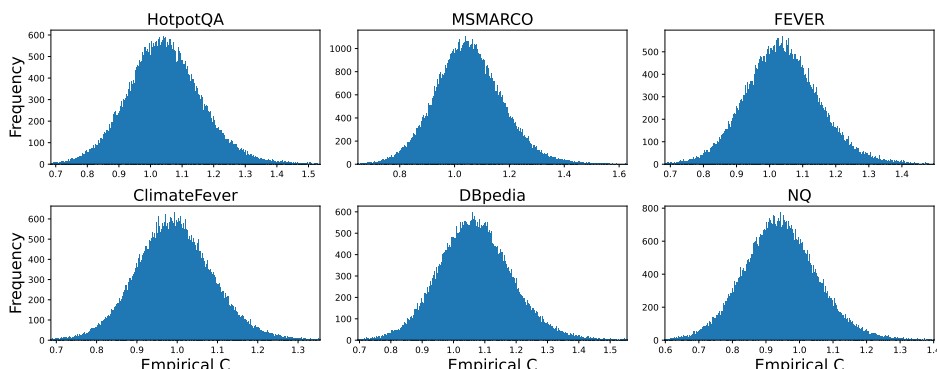

Figure 6: Results for 6 BEIR Retrieval datasets. Histograms of $C = D/d$ values, where we use "bge-micro-v2" as the distance proxy $d$ and "SFR-Embedding-Mistral" as the expensive distance $D$.

**Different nearest neighbor search algorithms**   We implement our method with another popular empirical nearest neighbor search algorithm called NSG Fu et al. (2019b). We obtain the same qualitative behavior as DiskANN. Because the authors' implementation of NSG only supports $\ell_2$ distances, we first normalize all the embeddings and search via $\ell_2$. This may cause some performance drops. Therefore, we are not comparing the results between the DiskANN and NSG algorithms, but only results from different methods, fixing the graph index. In Figure 12 and 13 in the appendix, we observe that our method still performs the best compared to Bi-metric (baseline) and single metric in most cases, demonstrating that our bi-metric framework can be applied to other graph-based nearest neighbor search algorithms as well.

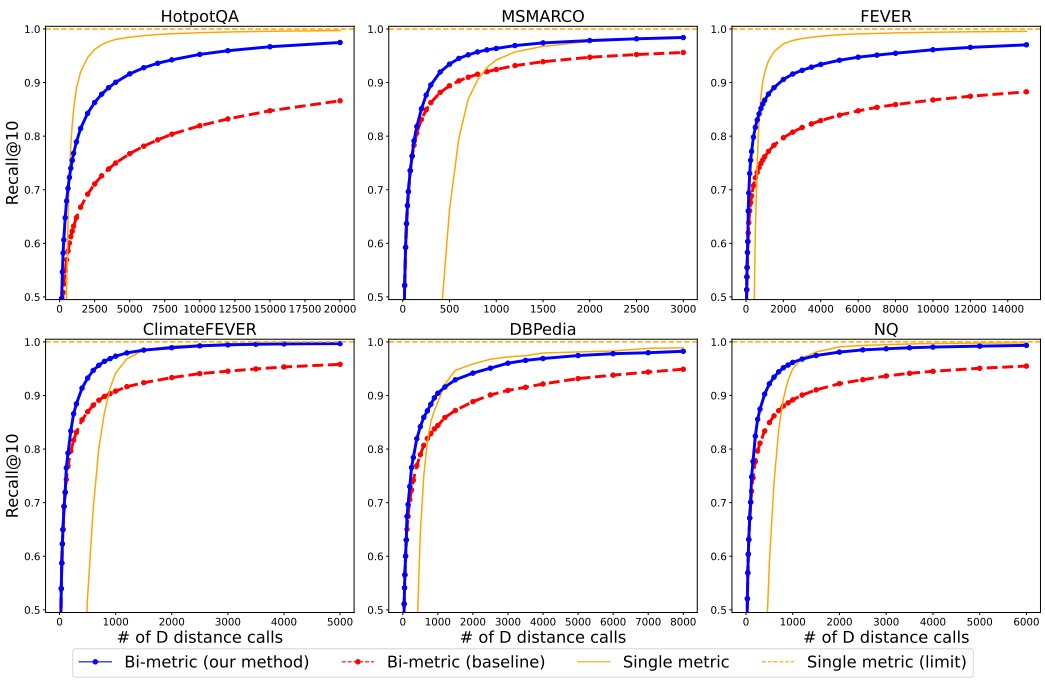

Figure 7: Results for 6 BEIR Retrieval datasets. The x-axis is the number of expensive distance function calls. The y-axis is the Recall@10 score. The cheap model is "bge-micro-v2", the expensive model is "SFR-Embedding-Mistral", and the nearest neighbor search algorithm used is DiskANN.

**Details of the LLM-based listwise reranking experiment**   Here we provide the details for our experiments in Section 4.2. Due to the fact that LLMs are better at comparing the relevancy of

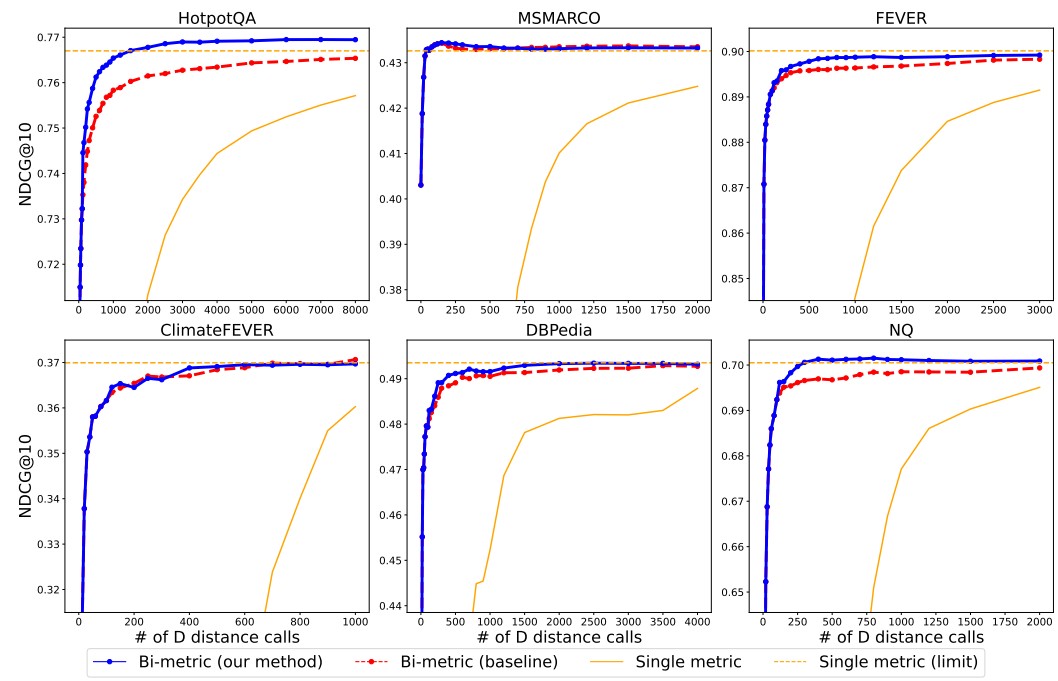

Figure 8: Results for 6 BEIR Retrieval datasets. The x-axis is the number of expensive distance function calls. The y-axis is the NDCG@10 score. The cheap model is "gte-small", the expensive model is "SFR-Embedding-Mistral", and the nearest neighbor search algorithm used is DiskANN.

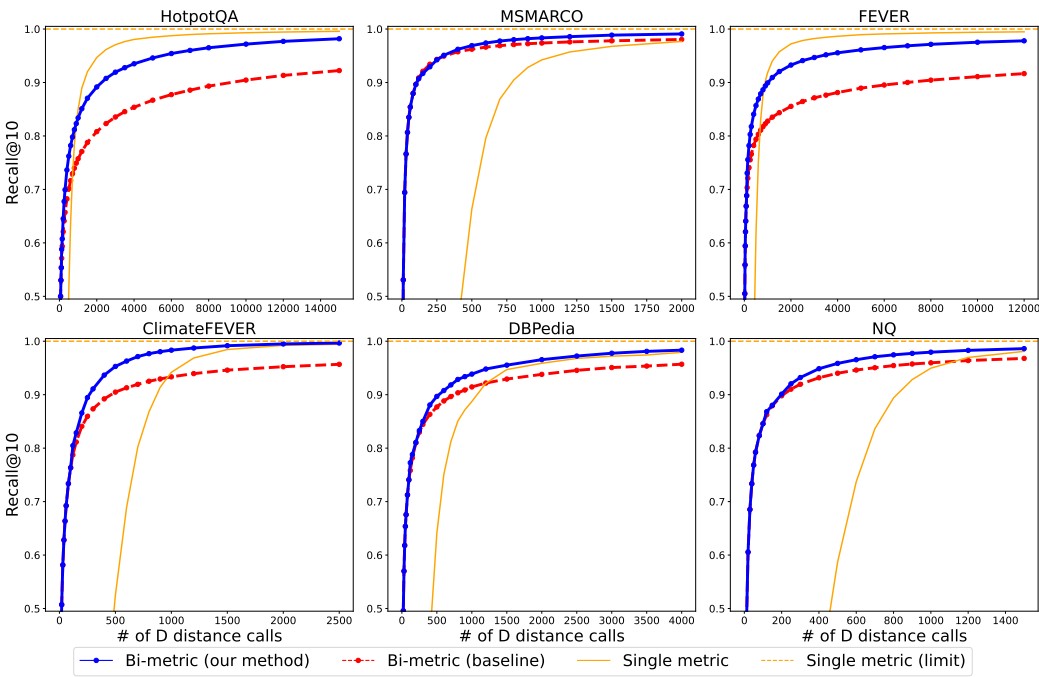

Figure 9: Results for 6 BEIR Retrieval datasets. The x-axis is the number of expensive distance function calls. The y-axis is the Recall@10 score. The cheap model is "gte-small", the expensive model is "SFR-Embedding-Mistral", and the nearest neighbor search algorithm used is DiskANN.

different passages than providing independent relevance scores, we need to modify our algorithm to maintain a list of $Ls$ current best answers. Please see our Algorithm 4. Its difference from Algorithm 1

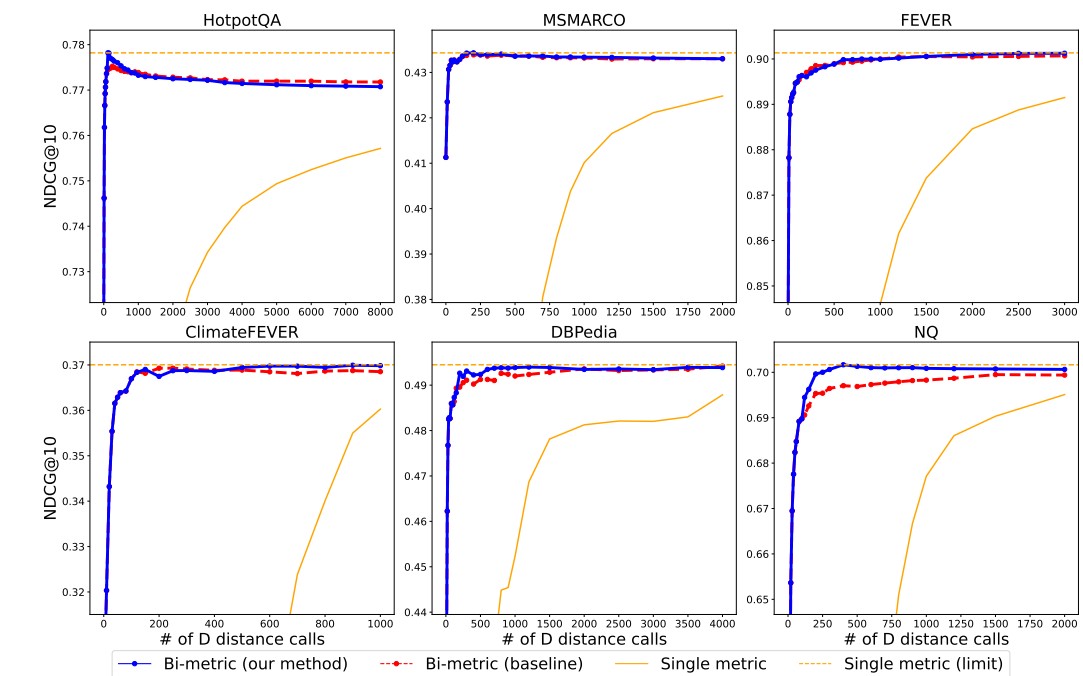

Figure 10: Results for 6 BEIR Retrieval datasets. The x-axis is the number of expensive distance function calls. The y-axis is the NDCG@10 score. The cheap model is "bge-base-en-v1.5", the expensive model is "SFR-Embedding-Mistral", and the nearest neighbor search algorithm used is DiskANN.

| System Instruction | You are RankGPT, an intelligent assistant that can rank answers based on their relevancy to the query. I will provide you with 10 passages, each indicated by number identifier []. Rank the answers based on their relevance to query: {query}. |
|---|---|
| Messages | [1] {Passage 1}
[2] {Passage 2}
...
[10] {Passage 10}
Query: {query}. Rank the 10 passages above based on their relevance to the query. The passages should be listed in descending order using identifiers. The most relevant passages should be listed first. The output format should be like [1] >[2] ... >[10]. Only response the ranking results, do not say any word or explain. |

Table 2: Prompt for "Gemini-2.0-Flash" to rerank passages

is that instead of maintaining a priority queue $A$, in Algorithm 4, $A$ is an ordered list. At each step, we first append all the unseen neighbors of $v$ to the end of $A$, and then perform a sequential reranking in a sliding window way to update the current top passages, similar to the application of listwise rerank in Sun et al. (2023). In the experiment, we set $Ls = 50$, $w = 10$. We start our second stage search from $max(50, Q/2)$ points retrieved by the first stage search where $Q$ is quota set to be the maximal number of passages seen by the reranker. Please See Table 2 for our prompt.

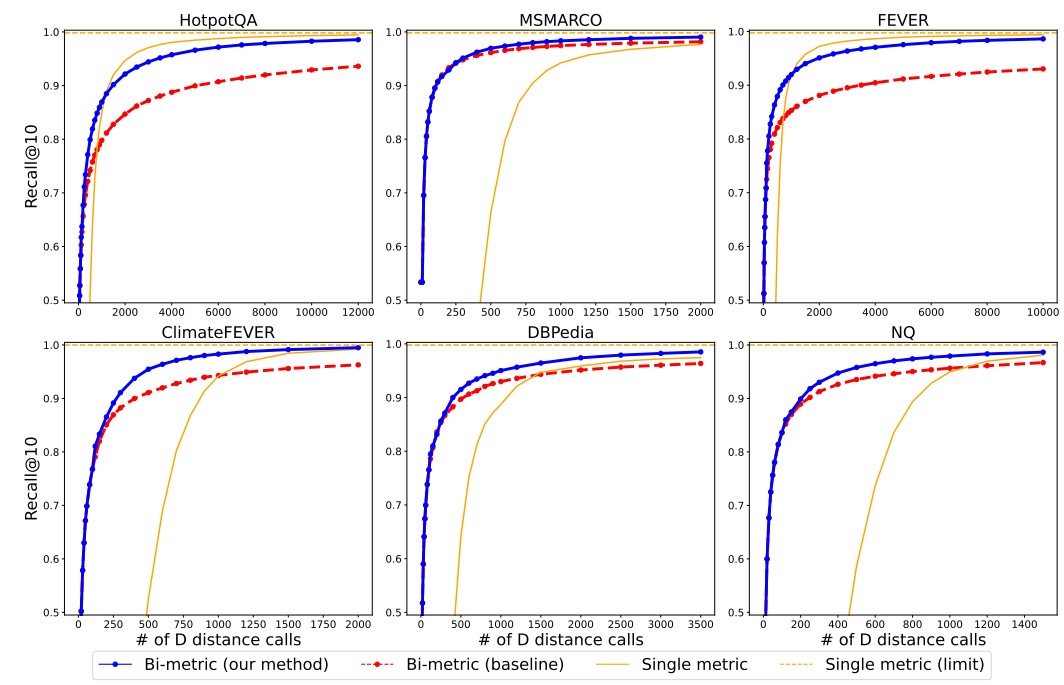

Figure 11: Results for 6 BEIR Retrieval datasets. The x-axis is the number of expensive distance function calls. The y-axis is the Recall@10 score. The cheap model is "bge-base-en-v1.5", the expensive model is "SFR-Embedding-Mistral", and the nearest neighbor search algorithm used is DiskANN.

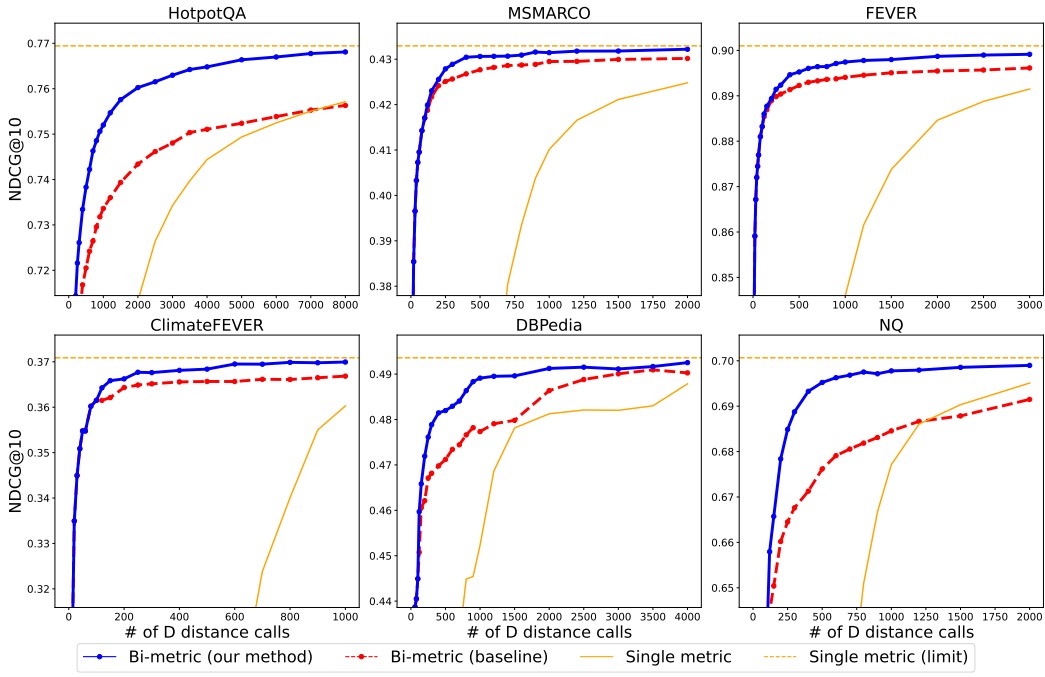

Figure 12: Results for 6 BEIR Retrieval datasets. The x-axis is the number of expensive distance function calls. The y-axis is the NDCG@10 score. The cheap model is "bge-micro-v2", the expensive model is "SFR-Embedding-Mistral", and the nearest neighbor search algorithm used is NSG.

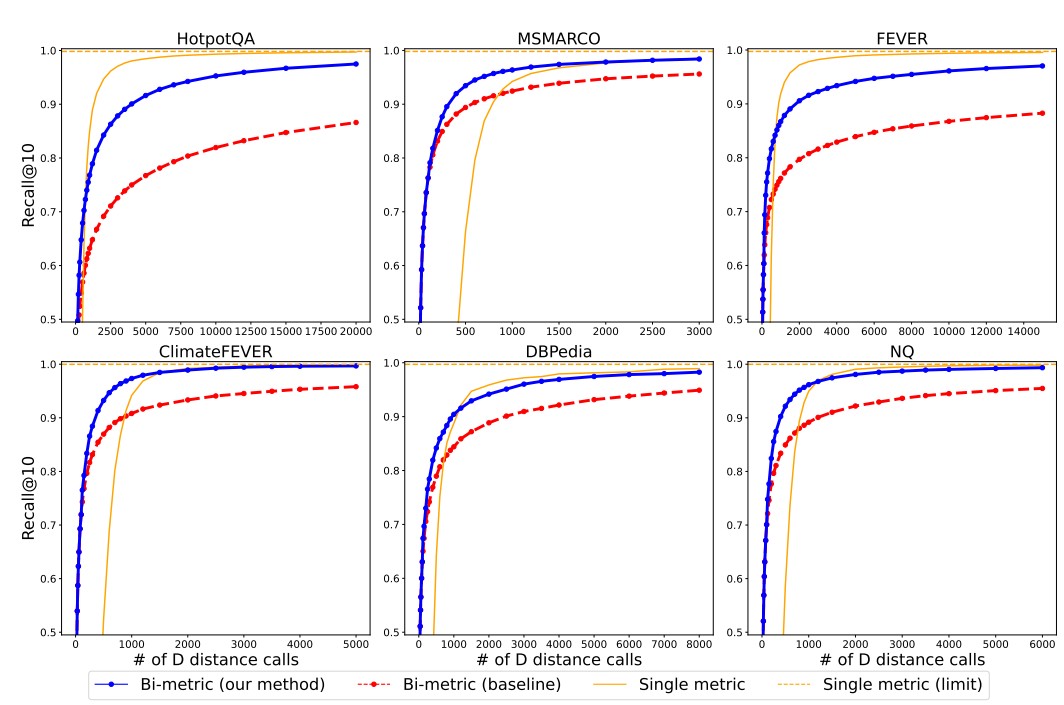

Figure 13: Results for 6 BEIR Retrieval datasets. The x-axis is the number of expensive distance function calls. The y-axis is the Recall@10 score. The cheap model is "bge-micro-v2", the expensive model is "SFR-Embedding-Mistral", and the nearest neighbor search algorithm used is NSG.

---

**Algorithm 4** Order-GreedySearch($q, d$)

---

1: **Input**: Graph index $G = (X, E)$, listwise-reranker $D$, starting point $s$, query point $q$, queue length limit $L$, sliding window size $w$.
2: **Output**: a sorted vertex list $A$
3: $A \leftarrow \{s\}$         ▷ An ordered list of vertices
4: $U \leftarrow \varnothing$
5: **while** $A \setminus U \neq \varnothing$ **do**
6:     $v \leftarrow$ the first vertex in $A \setminus U$
7:     $U \leftarrow U \cup v$
8:     Append $Neighbors(v) \setminus A$ to the end of $A$       ▷ Neighbors in $G$
9:     **for** $i = |A|$ to 0 step size $-w/2$ **do**
10:         Use D to rerank (A[$i - w$],···,A[$i$])
11:     **if** $|A| > L$ **then**
12:         $A \leftarrow$ the first $L$ vertices in A
13: **return** $A$

---

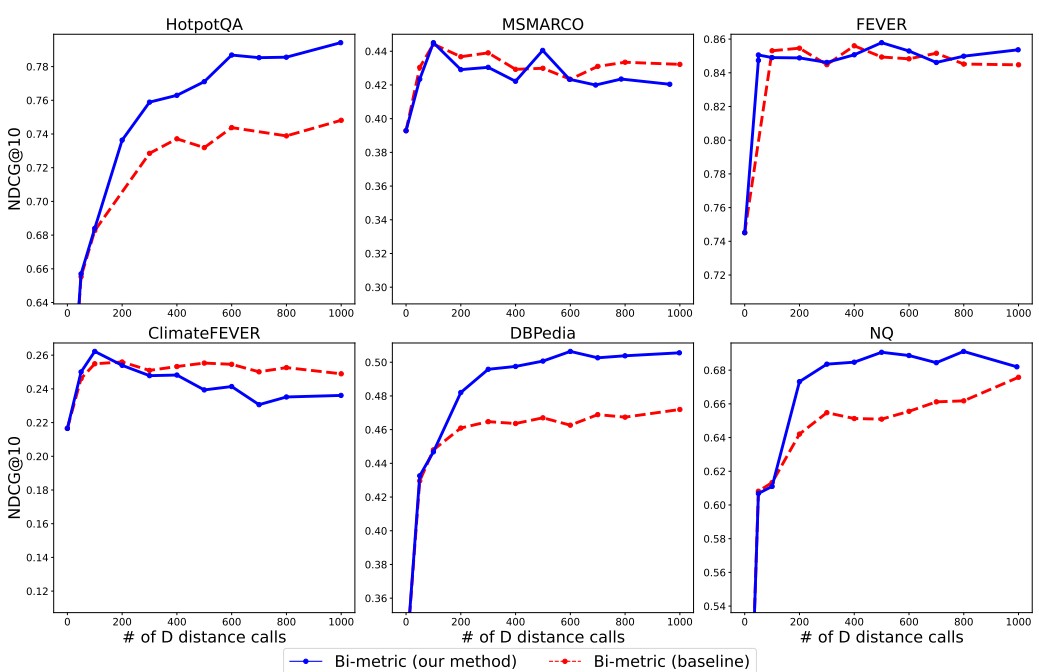

Figure 14: Results for 6 BEIR Retrieval datasets. The x-axis is the number of passages sent to the reranker. The y-axis is the NDCG@10 score. The cheap distance function is provided by "bge-micro-v2", the expensive model distance comparator is "Gemini-2.0-Flash", and the nearest neighbor search algorithm used is DiskANN.

# F DISCUSSION ON LSH

We give a very simple example, showing that the standard locality sensitive hashing (LSH) algorithm can be easily 'tricked' in our two distance functions setting, even if the distances approximate each other very well. Thus, locality sensitive hashing cannot be instantiated in the bi-metric framework, demonstrating the power of graph-based approaches. In fact, our construction extends to a broad class of algorithms which 'overfit' to the coordinates of the vectors. The intuitive idea is that any algorithm which 'only looks' at the coordinates of the query vector during the query phase can be fooled by fixing the coordinates of the query across the two metrics, but changing the coordinates of the input dataset slightly. At the core, LSH, as well as many partition based algorithms, can be abstracted into the following canonical form:

1. Given an input dataset $X \subset \mathbb{R}^d, |X| = n$ and an input metric, output a function $f : \mathbb{R}^d \to 2^{[n]}$.

2. Given a query point $q \in \mathbb{R}^d$, query $f(q)$ to return a subset of $[n]$.

3. Find the nearest neighbor of $q$ using the input metric among the points in $X$ whose indices are in $f(q)$. We define the running time of $f$ to be $O(|f(q)| \cdot T)$, where $T$ is the cost to evaluate the metric. For simplicity, we ignore this factor of $T$ in the subsequent discussion.

The main conceptual difference between the above canonical form and our graph based approach is that the query is used in 'one shot' to return the set $f(q)$ at once. This is problematic when using two distances in our bi-metric framework since $f(q)$ only depends on $d$ above (the cheap metric), but we want to find the nearest neighbor with respect to $D$ (the ground truth metric). In contrast, graph based approaches iteratively and adaptively build such a query set, based on the edges of the index graph and the corresponding search procedure on the graph. The fact that the query set is a function of *both* metrics in graph based search is crucial in avoiding the undesired behavior shown below.

We first note the following trivial observation.

**Observation:** Let $X, X' \subset \mathbb{R}^d, |X| = |X'|$ be two datasets with corresponding metrics $d$ and $d'$. Suppose we instantiate the above canonical algorithm on $X'$ using metric $d'$ and let $f'$ be the corresponding query function. Let $q$ be a query point for the dataset $X$ and let $i^*$ be the index of its nearest neighbor in $X$ with respect to $d$. If $i^* \notin f'(q)$ then we cannot find the nearest neighbor of $q$ in $X$ by only comparing the distances from $q$ to the points in $f'(q)$ (even if we evaluate using metric $d$ on the points in $f'(q)$).

Now lets discuss how the standard LSH algorithms fall in the above canonical description. We need to first specify a family of hash functions $\mathcal{H}$. Then for some suitably chosen parameters $k, L \geq 1$, we repeat the following procedure for $i = 1, \cdots, L$ iterations : Independently sample $k$ functions $h_1^i, \cdots h_k^i \sim \mathcal{H}$ and group the points by putting all $x$ with the same tuple $(h_1^i(x), \cdots, h_k^i(x))$ together. For a query $q$, retrieve all the points in $X$ with the same tuples $(h_1^i(q), \cdots, h_k^i(q))$ across all $i$. This forms the set $f(q)$. Usually, $|f(q)|$ is determined by the choice of $\mathcal{H}, k, L$ and for different metrics, researchers carefully choose these parameters to optimize for correctness and running time Andoni et al. (2018). We do not need these details in constructing the bad example and they can be abstracted away in our description of the canonical algorithm.

Our simple bad example for LSH is as follows. The dataset is $X = \{x_1, \cdots, x_n\} \subset \mathbb{R}^d$ (for $d$ sufficiently large). It consists of one copy of the first basis vector $x_1 = e_1$ and $n - 1$ copies of the sum of the first two basis vectors: $x_2 = \cdots = x_n = e_1 + e_2$. The ground truth metric $D$ will be the hamming distance on the vectors in $X$. The noisy metric $d$ will be the hamming distance on the corresponding points of a modified version of $X$. The modified dataset $X' = \{x_1', \cdots, x_n'\}$ is such that $x_1'$ is just $x_1$ but we set the last 10 coordinates to all 1's. For the other $x_i'$ vectors, $i \geq 2$, we keep the same $x_i$, but modify the last 5 coordinates to be all 1's. We have:

- $d$ approximates $D$ up to a factor of $O(1)$.
- The doubling dimensions of both $X'$ and $X$ (under $d$ and $D$ respectively) are $O(1)$.

We let the query vector $q$ be the all 0's vector ($q$ will be all 0's with respect to $D$ and $d$). In $X$, $x_1$ is the $c$-approximate nearest neighbor to $q$ for any $c < 2$. Note this is a setting where our theorems *guarantee* the performance of graph based algorithms (Theorems 3.3 and B.3), giving meaningful sublinear running time. For the rest of the section, we fix the query $q = 0$.

Now consider what happens when we use the standard hamming LSH function ($\mathcal{H}$ is the set of coordinate projection functions) and build a datastructure $f'$ using the noisy metric. Intuitively, unless $k$ and $L$ are sufficiently large, we cannot even guarantee with good probability that $1 \in f'(0)$ (note this means that the index of the first point is in the set $f'(0)$). However if we can guarantee that the 'correct' answer is in $f'(0)$, many irrelevant data points are also very likely to be in $f'(0)$, implying $|f'(0)| = \Omega(n)$, i.e. the query time is linear and hence not efficient. This is shown below.

**Lemma F.1.** *Suppose* $(k, L)$ *such that*

$$\Pr(1 \in f'(0)) \geq 0.01,$$

*i.e. the query is successful with probability at least* $1\%$. *With these same parameters, we have* $\mathbb{E}[|f'(0)|] = \Omega(n)$.

*Proof.* A simple calculation shows the following (since our hash functions are sampled from coordinate projections):

$$\Pr_{h \sim \mathcal{H}}(h(x_1') = h(0)) = \frac{d-11}{d} := p_1 \leq \Pr_{h \sim \mathcal{H}}(h(x_i') = h(0)) = \frac{d-7}{d} := p_2$$

for any other $i \geq 2$. If we repeat the hashing $k$ times, it is clear that the probability that $(h_1(x_1'), \cdots, h_k(x_1')) = (h_1(0), \cdots, h_k(0)) = p_1^k \leq p_2^k$. Thus, for any choice of $k$ and $L$, we have that for all $i \geq 2$,
$$\Pr(1 \in f'(0)) \leq \Pr(i \in f'(0)).$$
Hence, if $k, L$ is picked such that $\Pr(1 \in f'(0)) \geq 0.01$, it follows that

$$\mathbb{E}[|f'(0)|] = \sum_{i=1}^{n} \Pr(i \in f'(0)) \geq n \Pr(1 \in f'(0)) \geq \Omega(n),$$

as desired. □

In conclusion, the above lemma shows that unless the set $f'(q)$ is very large, which leads to a large running time for using LSH, it cannot be successfully used for nearest neighbor search with two metrics, in contrast to our graph based approach. The underlying idea of our bad example clearly generalizes across any reasonable choice of $\mathcal{H}$. The proof of the following corollary is identical to that of Lemma F.1.

**Corollary F.2.** *Suppose* $\mathcal{H}$ *is a family of functions with domain* $\mathbb{R}^d$ *such that* $\Pr_{h \sim \mathcal{H}}(h(x) = h(y))$ *is a decreasing function of* $\|x - y\|_2$. *Consider* $X$ *and* $X'$ *as defined above. Then if* $(k, L)$ *are picked such that*

$$\Pr(1 \in f'(0)) \geq 0.01,$$

*i.e. the query is successful with probability at least* $1\%$. *With these same parameters, we have* $\mathbb{E}[|f'(0)|] = \Omega(n)$.

The hypothesis on $\mathcal{H}$ in Corollary F.2 is quite natural and is satisfied by many natural choices. For example, the standard Euclidean LSH function class $\mathcal{H}$ consists of functions of the form $h_v(x) = \lfloor \langle x, v \rangle / a \rfloor$ where $v \sim \mathcal{N}(0, 1)$. A simple calculation shows that $\Pr_{h \sim \mathcal{H}}(h(x) = h(y)) \propto \|x - y\|_2$ and Corollary F.2 applies.

Upon a closer look, we can even abstract away all the details of LSH and return to the canonical form described in the beginning of the section. All we require for the bad example to hold is that $\Pr(i \in f'(0))$ is a decreasing function of $\|x_i' - q\|_2 = \|x_i'\|_2$.

**Corollary F.3.** *Suppose in our cannonical algorithm description that* $f'$ *is a function such that for all indices* $1 \leq i \leq n$, $\Pr(i \in f'(0))$ *is an decreasing function of* $\|x_i'\|_2$. *If*

$$\Pr(1 \in f'(0)) \geq 0.01,$$

*i.e. the query is successful with probability at least* $1\%$, *then we also have* $\mathbb{E}[|f'(0)|] = \Omega(n)$.

## G USAGE OF LARGE LANGUAGE MODELS

As mentioned in Section 4.2, we apply an LLM-based reranker in our experiments. We also use LLMs to polish writing.

