# OpenReview forum: "A Bi-metric Framework for Efficient Nearest Neighbor Search"
_ICLR.cc/2026/Conference — Submitted to ICLR 2026_

### Official Review · Reviewer_GeBN · 2025-10-16

**Soundness:** 3
**Presentation:** 3
**Contribution:** 1
**Rating:** 2
**Confidence:** 4

**Summary:**

The problem the paper considers an approximate nearest neighbor search under an expensive-to-compute dissimilarity measure $D$ with the assumption that there exists a cheaper dissimilarity measure $d$ that approximates it. The task is to find the approximate nearest neighbors (w.r.t. $D$) for the query point $q \in \mathbb{R}^d$ from the data set $P \subset \mathbb{R}^d$ within a budget $\mathcal{Q}$ of evaluations of the expensive distance metric $D$ (the number of evaluations of $d$ is unlimited). In the experiments of the paper,  $D$ is instantiated by the Euclidean distance between the embeddings computed by large and expensive-to-compute bidirectional encoder, and $d$ by the Euclidean distance between the embeddings computed by a smaller model. A traditional approach to this task is "retrieve-then-rerank", where $\mathcal{Q}$ neighbors of $q$ under $d$ are first retrieved, and then these $\mathcal{Q}$ points are reranked by evaluating $D$ between them and the query point.

The article proposes an approach where a graph-based index is built in the offline phase using a proxy metric $d$. Then in the query phase, $\mathcal{Q} / 2$ nearest neighbors are retrieved using $d$. Then, these $\mathcal{Q} / 2$ are used to initialize greedy search in a graph using $D$. The search terminates when $\mathcal{Q}$ distances are evaluated.

The experimental results of the article demonstrate that the proposed approach (when combined with graph indexes DiskANN and NSG) enables reaching a higher recall with a same budget $\mathcal{Q}$ than retrieve-then-rerank approach. The article also provides theoretical guarantees for their approach when combined with DiskANN and cover tree indexes.

**Strengths:**

(S1) The experiments of the article demonstrate that the proposed approach outperforms retrieve-then-rerank under the assumptions of the cost model of the article.

(S2) Theoretical guarantees are provide for the approach (when combined with DiskANN and cover tree indexes).

**Weaknesses:**

(W1) Methodological novelty of the article is very limited. As stated in the summary, the proposed "framework" is: 1) Build a graph index using a proxy metric $d$ in the offline phase 2) In the query phase, first find the $\mathcal{Q}/2$ neighbors of the query point under $d$. 3) Initialize greedy search from these points using $D$, and continue greedy search until $\mathcal{Q}$ distances are evaluated. On a related note, "Bi-metric framework" on page 4 (lines 180-194) is not a framework, but a problem definition.

(W2) The cost model used to motivate the approach and used in the experiments is not realistic in real-world applications (or, more exactly is realistic only on a very niche scenario mentioned by authors where the expensive model used to evaluate $D$ may change very often). Specifically, the authors assume that the embeddings of the expensive model (used to evaluate $D$) are not precomputed, but are always computed on demand when performing queries. If these embeddings were precomputed, queries would be much faster compared to the experimental scenarios presented in the paper, making the experiments irrelevant. The authors motivate their assumption that the embeddings of the large model are not precomputed by stating that they are expensive to compute. However, if the index is used to answer any number of queries that is realistic in an industrial application, almost all of the embeddings of the points of the data set $P$ have to computed anyway when the queries are answered. Thus, they could be as well precomputed (or cached from the previous queries).

**Questions:**

I do not have any particular questions for the authors.

---

> ### Author Response · Authors · 2025-11-22
>
> Reply to W1: Indeed our method is simple to implement, but it beats the standard retrieve-than-rerank baseline across our experimental settings. Furthermore, we give worst-case theoretical guarantees of our method based on the doubling dimension parameterization, whereas the retrieve-and-rerank baseline has no meaningful theoretical guarantees. Thus, we believe our result achieves a ‘best-of-both worlds’ scenario and its clean approach and theoretical guarantees are strengths of the method.
>
> Reply to W2: We believe this comment is not correct. For example, our experiments with Gemini model (cf. Figure 2) does not involve embeddings at all. Instead, the expensive metric evaluations are implemented using calls to a proprietary model, so the overall search cost (expressed in dollars) is directly proportional to the number of expensive metric evaluations.

---

> > ### Comment · Reviewer_GeBN · 2025-11-27
> >
> > You are correct that scenario where a LLM is used as a reranker is a realistic use case of your framework. However, the main experimental results of your article are performed in the setting where a larger embedding model is used to compute the expensive metric, and for the reasons stated in my original review, I do not think that is a practical use case. Also observe that in the LLM as reranker experiment, your method improves over the baseline only on 3 out of 6 data sets.
> >
> > However, I increase my score to 4 since the LLM as reranker use case is relevant.

---

### Official Review · Reviewer_aQro · 2025-10-31

**Soundness:** 2
**Presentation:** 4
**Contribution:** 2
**Rating:** 4
**Confidence:** 4

**Summary:**

The authors consider a scenario for approximate nearest neighbor search where two metrics are involved: a cheap metric that is used to construct the ANN index, and an expensive metric with respect to which the nearest neighbors are found. A common instantiation of this scenario is to first retrieve a larger candidate set of nearest neighbors using the cheap metric and then rerank those points using the expensive distance metric.

The authors prove that under certain assumptions, the DiskANN and Cover Tree algorithms can be used to build a data structure using the cheap metric such that they can be used to query for the nearest neighbors using the expensive metric with only a sublinear number of expensive metric evaluations. A practical application of the proposed method is simple: first construct a graph index using the cheap metric. In the query phase, given a query budget $Q$ of calls to the expensive metric, first retrieve $Q/2$ nodes in the graph using the cheap distance metric, and then use those nodes as seed points for performing $Q$ distance computations using the expensive metric. The authors perform experiments where the cheap metric is a small embedding model, and the expensive metric is a large embedding model or an LLM listwise reranker.

**Strengths:**

The article addresses an interesting and, to the best of my understanding, novel topic. The proposed method is theoretically motivated and in practice, graph-based algorithms (which are demonstrated using DiskANN and NSG) are straightforward to modify for the proposed bi-metric framework. The experiments show that the method can yield improved results in certain cases. The paper is well written and easy to read.

**Weaknesses:**

While I like the idea conceptually and the theory involved in the framework is nice, I think the experiments are not convincing enough to indicate that the framework or method would have practical impact.

In particular, as the expensive distance metric $D$, the authors consider only two choices, a 7 billion parameter embedding model and an LLM listwise reranker. The large embedding model is a somewhat unrealistic choice: the whole point of using an embedding model is that you need to compute the embedding only for the query during retrieval. Calling a 7B embedding model hundreds of times during retrieval is infeasible (I understand the indexing time can also be infeasible but it is a secondary concern).

Similarly, when using an LLM as a reranker you would typically only rerank a maximum of a few dozen queries due to the speed and costs involved. Moreover, for the listwise reranker the method only works better on half the datasets, and the experiments use a proprietary model.

A natural choice would be of course to instead consider a cross-encoder as $D$. There are many available open-source cross-encoder models that would be quite suitable at 100M-1B parameters, and it is unclear why the authors do not consider these at all. If the authors also want a listwise approach, there are suitable rerankers available for that as well, e.g. the recent jina-reranker-v3 [1].

Finally, of course an advantage of the simple retrieve-then-rerank approach is that it is easy to combine the retrieve phase with e.g. sparse search or filtering, and only rerank those results.

[1] F. Wang, Y. Li, H. Xiao. jina-reranker-v3: Last but Not Late Interaction for Listwise Document Reranking. arXiv:2509.25085. 2025.

**Questions:**

- Could the authors clarify reasons or propose a hypothesis for why the "single metric" method performs so poorly?

- Have you tried how your method works with other graph methods such as HNSW?

- Is source code available for reproducability?

---

> ### Author Response · Authors · 2025-11-22
>
> Regarding “open-sourced cross-encoders”:  Thanks for your suggestion! We have done preliminary experiments with cross-encoders before. However, we believe that there is still a substantial gap between those trained cross-encoders and general LLM models as reranker models. Specifically, those cross-encoder models won’t help retrieval quality when the number of scanned documents is over a few dozen. There is a research paper summarizing this phenomenon [1].
>
> Regarding the embedding model: indeed, computing the embeddings “on-the-fly” during the query phase is expensive. However, the advantage of this setting is that we could perform the experiments with only a limited compute budget (unlike our second experiment with Gemini, where the cost of accessing the model was quite severe). Furthermore, it makes it possible to use different metrics D without recomputing all embeddings from scratch.
>
> Reply to Q1: It’s not that “single metric” performs poorly. It’s actually pretty fast to reach the optimal recall using only a few thousand of distance computations on a million-scale dataset. Our bi-metric method is able to improve upon that because we can get access to a much cheaper distance “proxy” which approximates the expensive metric very well, which enables us to use even fewer expensive distance computations.
>
> Reply to Q2: We have also performed experiments with the NSG algorithm (Figures 11 and 12). We did not try HNSW, though we expect the results to be similar.
>
> Reply to Q3: Yes, we have released our code online. Unfortunately, for anonymity reasons, we can’t include the link here.
>
> [1] Jacob, M., Lindgren, E., Zaharia, M., Carbin, M., Khattab, O., & Drozdov, A. (2024). Drowning in documents: Consequences of scaling reranker inference. Preprint.

---

> > ### Comment · Reviewer_aQro · 2025-11-27
> >
> > Thank you for your reply.
> >
> > It is unfortunate that cross-encoders do not work well in this setting as this would have been a clear potential application of your method.
> >
> > > Regarding the embedding model: indeed, computing the embeddings “on-the-fly” during the query phase is expensive. However, the advantage of this setting is that we could perform the experiments with only a limited compute budget (unlike our second experiment with Gemini, where the cost of accessing the model was quite severe). Furthermore, it makes it possible to use different metrics D without recomputing all embeddings from scratch.
> >
> > I acknowledge these reasons but they do not really change the fact that the setting is unrealistic (e.g. the possibility to switch $D$ doesn't really matter if this method would never be used in practice to begin with).
> >
> > Using the LLM as a reranker has more potential but as mentioned in my original review, this also suffers partly from the same issue and the experiments use only a single proprietary model where the proposed method works better only half the time. I acknowledge that experimenting with additional models would be difficult with a limited budget but unfortunately in its current state the paper doesn't offer enough indication of potential impact for me to support acceptance.
> >
> > > Yes, we have released our code online. Unfortunately, for anonymity reasons, we can’t include the link here.
> >
> > I fully trust that the code will be available, but for the future do note that there exist many avenues for anonymous code release.

---

> > > ### Comment · Reviewer_SDf3 · 2025-11-27
> > > **agree**
> > >
> > > BTW, I also agree that using such expensive models directly for the 2d stage is a bit unrealistic. The gains are already pretty limited and they would probably shrink further if the 2d stage model were an intermediate model. In fact, as I pointed out, many production-level systems have multiple re-ranking stages.
> > >
> > > That said, does this invalidate the science in the paper? Not really, but it can be admitted as a core limitation.

---

### Official Review · Reviewer_hixp · 2025-11-02

**Soundness:** 2
**Presentation:** 2
**Contribution:** 2
**Rating:** 4
**Confidence:** 3

**Summary:**

The author proposes a data structure design framework for nearest neighbor search: constructing an index using cheap but less accurate proxy metrics and performing only a few expensive and more accurate metric calls during the query phase. This approach achieves retrieval accuracy close to that of expensive metrics, while keeping the overall cost acceptable. In the preprocessing phase, the index is built using only the proxy metrics, while during the query phase, expensive metrics are used to perform greedy or hierarchical searches along the graph. The author shows that this separation of construction and querying approach still reports reliable nearest neighbor results under expensive metrics.
Key Contributions:
1. Proposed dual-metric framework: The index is built using approximate metrics, while the query phase combines real metrics to balance computational efficiency and accuracy.
2. Theoretical proof: For two mainstream algorithms—DiskANN and Cover Tree. The author proves that as long as the deviation between the approximate and real metrics is bounded, a theoretical guarantee of approximation close to the real metric can be obtained.
3. Experimental validation: The framework is validated in text retrieval tasks (BEIR dataset), where it outperforms the traditional "retrieve-then-rerank" method in accuracy-efficiency trade-offs on almost all large-scale datasets.

**Strengths:**

1. The balance between accuracy and efficiency in NN problem is interesting. The motivation is good.
2. The author proposes a general dual-metric retrieval framework. In this framework, the author constructs the index with cheap approximate metrics and performs only a few calls to expensive real metrics during the query phase, while maintaining results close to those of the real metrics.
3. The paper provides a systematic theoretical analysis, showing that building an index using only approximate metrics and using real metrics during the query phase can still yield high-quality results. It also presents comparisons and counterexamples, explaining why methods like LSH are not suitable for this framework.

**Weaknesses:**

1. The method, both theoretically and experimentally, relies on the assumption that the difference between the cheap and real metrics is bounded, but the samples and domains are relatively narrow (text retrieval).
2. To ensure recall during the cheap-metric phase, the authors set a very long query length for graph search, which may favor the proposed method. A systematic study of hyperparameter sensitivity and ablation (e.g., varying query lengths, graph construction strategies) is recommended.
3. When using LLMs for list-wise reranking, the authors acknowledge that occasional ranking errors cause fluctuations in the tail of the curves. Currently, only average trends are shown.
4. The experiments are costly. For example, reranking experiments costs thousands of dollars, making it difficult for other researchers to reproduce the results.
5. The structure needs optimization to make it more easy to follow. Some mistakes are made. e.g., The period is forgotten (First paragraph in Introduction section). The comma is missed (The second to last paragraph on Page 2). The proxy metric d and the data dimensionality d are reused.

**Questions:**

1. The paper assumes a bounded distortion relationship between the cheap metric d and the expensive metric D. In real-world applications, if the ratio D/d deviates significantly (e.g., C>2 or is unstable), how would the algorithm's performance change?
2. The paper mentions that to ensure sufficient recall in the first phase, a very large query length (e.g., L=30000) is used. Has the author tested more conventional parameter settings? How sensitive is the method's performance to these parameter changes? It would be helpful for the author to provide hyperparameter sensitivity or ablation experiments to assess the method's stability and ensure fair comparison.
3. The paper primarily measures efficiency in terms of the number of expensive metric calls, but does not report end-to-end runtime or other efficiency metrics. Could the author provide the overall time cost, including both index construction and the query phase?
4. What will the framework perform in image retrieval or cross-modal retrieval tasks (two typical NN search applications)?

---

> ### Author Response · Authors · 2025-11-22
>
> Reply to W1 and Q1: Indeed, our theoretical bounds need the gap C between the cheap and expensive metrics to be bounded. Unfortunately, this is unavoidable – if the two metrics are unrelated, then the data structure developed for the cheap metric provides no help, and a linear scan is required in the second stage. However, our empirical algorithms and experiments do not require C to be bounded, at least not explicitly. (Though if the two metrics are not related, the second stage search will devolve into a linear scan, as in theory, eliminating any benefits of our method.)
>
> We believe that text retrieval is a highly suitable testbed for our algorithm, as it has widely used benchmarks and a large collection of different models with varying sizes. Also, as per Figure 6, our selection of cheap and expensive metrics yields a small ratio C=D/d
>
> Reply to W2 and Q2: Thanks for your suggestion. We have conducted ablation studies to evaluate the influence of the first-stage queue length $L$ on NQ and HotpotQA. Please see the results in Figure 5 in the appendix. In short, the sensitivity with respect to the first-stage search queue length $L$ varies across different datasets. Generally, a much smaller $L = 1000$ suffices. Our use of an extremely large $L = 30000$ is intended to approximate brute-force retrieval, rather than being strictly necessary.
>
>
>
>
> Reply to W3: Please see full Gemini experimental results, which are already included in Figure 14.
>
> Reply to W4: Indeed, our experiment with the proprietary Gemini model was quite expensive for our budget.  However, we have also performed experiments on open-source embedding models, as per Figure 1. Those experiments were cheaper, though still requiring 196 GPU-hours on an A100 GPU.
>
> Reply to W5: Thank you for your suggestions! We have implemented your writing suggestions in the updated version.
>
> Reply to Q3: We focused on the number of expensive distance computations as the cost metric because in many scenarios, the cost is expressed in dollars, not time. In particular, this is the case in our Gemini experiments, where the total cost is proportional to the number of API calls needed to perform the expensive distance computations. For our experiments with embedding models, the query cost depends on whether the embeddings are computed on-the-fly or pre-computed in advance. If the embeddings are computed on-the-fly, we provide the timing information on page 8 (in short, the cost of invoking the expensive metric D dominates the query time). If embeddings are precomputed in advance, the costs of evaluating d and D are comparable, so there is no need to use our framework
>
> For the indexing phase, we are using the standard DiskANN algorithm, which means that all methods have the same indexing time. For example, it takes 570s to build the index on HotpotQA (5.23M) and 202s to build the index on NQ (2.68M)
>
> Reply to Q4: Thank you for this idea. Given our theoretical guarantees over the standard retrieve-and-rerank baseline and strong empirical performance across a wide range of text datasets, we believe our method can generalize to any setting where a cheap approximation to the true underlying similarity is available. If the reviewer has any specific suggestions of state-of-the-art multi-modal models that can be experimented with (on a limited budget), we will investigate this further.

---

### Official Review · Reviewer_SDf3 · 2025-11-03

**Soundness:** 3
**Presentation:** 3
**Contribution:** 3
**Rating:** 8
**Confidence:** 4

**Summary:**

This paper introduces a **bi-metric framework** for efficient nearest-neighbor search that combines a cheap, approximate metric with an expensive, accurate one. The method builds an index using only the proxy distance and performs search using both distances.

 Theoretical analysis provides guarantees for **DiskANN** and **CoverTree**, demonstrating that under a bounded distortion between metrics, the resulting data structures can approximate the results of the expensive metric.

Empirically, the authors validate the approach on large-scale **BEIR** retrieval datasets, pairing lightweight embedding models such as bge-micro-v2 with stronger ones like SFR-Embedding-Mistral or even LLM-based comparators such as Gemini-2.0-Flash. Across benchmarks, the bi-metric method achieves the same accuracy as traditional re-ranking while requiring **3–4× fewer evaluations** of the expensive model. This efficiency gain translates into non-trivial reductions in time and cost, and the framework remains effective even when the second “metric” is a **non-metric** listwise reranker.

Overall, the paper offers a **simple, general, and well-theorized alternative** to retrieve-and-rerank pipelines. It highlights that guiding search with a cheap proxy distance---then carrying a "tail" search using the expensive metric---can improve scalability with only modest decrease in quality compared to running an (often impractically expensive) search using solely the expensive metric.

**Strengths:**

1. A simple yet effective idea of carrying out a tail search: applying a computationally expensive distance only after obtaining a seed set of candidates through a cheap search with a proxy distance.

2. The approach is generic and applicable to multiple retrieval algorithms!

3. For two algorithms (graph-based retrieval and CoverTrees) it even comes with theoretical guarantees!

4. What is interesting the second “metric”, in practice, actually does not have to be the metric (although the theory will not hold).

5. Solid experimental results

6. The paper is generally well written although some improvements in clarity can be made (in particular naming of baselines is not very informative)

**Weaknesses:**

1. Presentation can be improved (see some detailed notes).



2. Compared to retrieve-and-rerank, gains are somewhat limited in the case of human-labels and NDCG@10. Although in L396 authors notice that gains are more substantial when measured through Recall@10. However, they do it in passing and it may be a lost opportunity to “sell” their approach properly. The issue with benchmarks like BEIR is that human labels are sparse and biased toward lexical search methods (since many annotations were generated by running lexical retrieval and labeling the top-K results). As a result, these benchmarks are likely not very sensitive to recall improvements from vector (semantic) search: once a method retrieves most of the human-labeled relevant documents, its score stops improving: even if it continues to find additional relevant items that were never labeled
3. An actual search engine can use more than one re-ranking stage. For example, you can retrieve top-1000 using a very cheap embedding model, rescore them using an embedding model with a medium cost and finally apply your expensive model to top-100 results. That said, I would consider this only as an optional baseline and one can always argue that intermediate rankers add extra complexity.

**Some editing recommendations**


LL113-115 Here, we use the fact that graph-based algorithms for nearest neighbor search do not require the values D(p, q) per se, but only use comparisons between D(q, p) and D(q, s). -> During retrieval the model keeps a priority-queue of M closest neighbors. When new candidates come in, they need to be merged with existing candidates. This can be sort of inefficient. In any case, it’s good to refer here the reader here to Algorithm 4 in the Appendix. It would have saved me some time.

L134-138  A good “classic” reference here is Matveeva, Irina, et al. "High accuracy retrieval with multiple nested ranker." Proceedings of the 29th annual international ACM SIGIR conference on Research and development in information retrieval. 2006.


L159 It might be worth mentioning (though **definitely not required**) that before Morozov and Babenko, Boytsov explored indexing and retrieval with different similarity measures in the context of graph algorithms and VP-trees. In one case, a cheap inner product was used to construct the k-NN graph, while retrieval relied on a more expensive similarity. This earlier work did not include theoretical guarantees and showed improvements over filter-and-rerank on one dataset out of three..

Boytsov, Leonid. Efficient and accurate non-metric k-NN search with applications to text matching (**Section 3**). Ph. D. Thesis. Carnegie Mellon University, 2018.

L239 For a metric d, ∆d is the aspect ratio of the input set: the ratio between the diameter and closest pair -> This needs to be a definition. Moreover, it needs to be rewritten for clarity. You probably mean the ratio between the diameter (what is this exactly, distance between two farthest points) and the distance between closest pairs.

L240 Note that both Definition 2.1 and 2.2 are only theoretical analysis: Definitions are not theoretical analysis.

L241 Our experimental results verify the advantage of our bi-metric framework without any assumptions; see Section 4. -> It’s not clear what you mean by “advantages without assumptions”.

L327 We help this method and ignore the large number of D distance calls in the indexing phase and only count towards the quota in the search phase. -> Please rewrite, this is a very awkward way to state that you use the expensive distance during the indexing phase as well. However, during retrieval you apparently still use your two-step algorithm.

L370-372 This is rather confusing because you compute recall as k-NN recall, i.e., use true k-nearest neighbors as relevant entries. However, for NDCG@10 you use human-labeled ground truth entries. This needs to be clarified. It is possible to compute recall with respect to human labels as well!

Bi-metric baseline is an extremely confusing name: please, use retrieve-and-re-rank instead

Single-metric is somewhat confusing as well because it doesn’t reflect the fact that the retrieval is budgeted. Maybe something like single-expensive-metric-with-query-budget (or some compressed version of it will be more natural).

Matveeva, Irina, et al. "High accuracy retrieval with multiple nested ranker." Proceedings of the 29th annual international ACM SIGIR conference on Research and development in information retrieval. 2006.

**Questions:**

Algorithm 1 and 4 are attributed to DiskANN: What is DiskANN specifics here? I think this is a standard 1-greedy retrieval algorithm that is known literally for dozens of years. Is the difference in the indexing algorithm? It would be nice to separate the indexing and retrieval algorithms.

---

> ### Author Response · Authors · 2025-11-22
>
> Reply to W1: Thank you for the writing suggestions. We implemented most of them in the current version. We will replace the term “baseline metric” in the final version, as this requires redoing the plots from scratch.
>
> Reply to W2: We agree with your observation. Our methods perform even better on standard recall metrics. According to [1], there is some noise in the BEIR benchmark, e.g., there are unlabeled relevant documents, which means that even if the algorithm finds new relevant documents, it may not be reflected in NDCG improvement. That may explain why our algorithm’s improvement is higher on the recall metric, where the ground truth is measured according to the expensive model. The reason we focused on NDCG despite its flaws is that this is a more popular metric for measuring retrieval accuracy.
>
> Reply to W3: Indeed,  a search engine can have more than one re-ranking stage. The reason we focused on two stages is that it separates the first stage, where we can use an index to speed up the retrieval, from the second stage, which only performs filtering. (It is not possible to use an index in the second stage, as the set of points retrieved in the first stage depends on the query). However, one can indeed expand the second stage into multiple re-ranking stages.
>
> Reply to Q1: Algorithm 1 is a generic greedy search, included for completeness. Algorithm 4 is also not tied to a specific indexing method, but since there is no agreement now on how to apply re-rankers to greedy search, we specified the details of our method.
>
> [1] Weller, Orion, et al. "Rank1: Test-time compute for reranking in information retrieval." arXiv preprint arXiv:2502.18418 (2025).

---

### Official Review · Reviewer_JPva · 2025-11-05

**Soundness:** 4
**Presentation:** 4
**Contribution:** 3
**Rating:** 8
**Confidence:** 4

**Summary:**

The paper introduces a "new" bi-metric framework for approximate nearest neighbor (ANN) search that operates over two dissimilarity functions:
(1) an expensive but accurate metric $D$ and
(2) a cheap yet approximate proxy metric $d$.
The key idea is to build the ANN data structure using only the proxy metric to ensure efficiency, while maintaining the accuracy of the true metric $D$ during query time through a sublinear number of calls to both metrics.

The authors instantiate this framework on two well-known graph-based algorithms, DiskANN and Cover Tree, and provide theoretical guarantees showing that if the proxy metric approximates $D$ within a bounded factor $C$, the structure achieves arbitrarily tight $(1 + \varepsilon)$ approximation under $D$.

Extensive experiments on six large-scale BEIR benchmark datasets demonstrate that the proposed method reduces expensive model evaluations by up to $4\times$ compared to standard retrieve-then-rerank pipelines, while matching or surpassing their accuracy.

**Strengths:**

- Introduces a semi novel bi-metric (refer to questions for why semi) framework that combines a cheap proxy metric with an expensive ground-truth metric, supported by solid theoretical guarantees.
- Provides formal proofs of sublinear query complexity and $(1+\varepsilon)$ approximation, improving upon standard retrieve-then-rerank approaches.
- Extends theoretical analysis to DiskANN and Cover Tree, and clearly delineates the limitations for LSH-based methods.
- Addresses a practical efficiency–accuracy tradeoff in large-scale retrieval, modeling computational and monetary costs realistically.
- Validates the framework empirically on six BEIR datasets, showing up to $4\times$ reduction in expensive model calls.

**Weaknesses:**

- The authors briefly acknowledge that the proposed bi-metric framework relies heavily on the proxy metric $d$ being a reasonably good approximation of the ground-truth metric $D$.
In the conclusion, they state:
``Our framework requires that the (cheap) proxy metric $d$ provides a ‘reasonable’ approximation to the (expensive) ground-truth metric $D$... performance may degrade if the proxy metric is a poor approximation of the ground truth.''

- They further note that this limitation is inherent when relying on a proxy, since in the extreme case where $d$ provides no useful information about $D$, the framework cannot yield accurate results.

- Empirically, the paper supports this assumption by reporting that the ratio $C = D/d$ is tightly concentrated around 1 (e.g., $0.6 \le C \le 1.5$ for HotpotQA), indicating strong alignment between the proxy and ground-truth metrics. However, this analysis only covers well-aligned embedding models (e.g., bge-micro-v2 vs. SFR-Embedding-Mistral) and does not explore failure cases.

- The paper does not provide:
1.  A systematic study of when or why the proxy metric fails (e.g., under domain shift or weak semantic correlation).
2. Insight into experiments with poor or random proxies.
3. Theoretical sensitivity or robustness bounds for large distortion constants $C$.

**Questions:**

1. The main theorem assumes $d$ closely approximates $D$. How realistic is this in practice? Please provide a real-world example where this assumption might fail.

2. The thoery focuses on DiskANN and Cover Tree, exploiting their graph and net structures. Could similar guarantees be extended to other graph-based methods such as HNSW or NSG, or are there fundamental barriers preventing this?

3. The analysis depends on bounded doubling dimension $\lambda_d$ for space and query complexity bounds. How sensitive are these guarantees when this assumption fails, as often occurs in embeddings used for text or vision?

4. The comparison between the proposed method and the bi-metric baseline is based on a fixed $D$-call budget $Q$. How were the choices of $Q/2$ and $Q$ determined, and could this parameterization influence the observed efficiency gap?

5. All experiments are conducted on BEIR text datasets with embedding and LLM-based metrics. How would the framework perform on other modalities (e.g., image or multimodal retrieval) or with weaker proxies where the distortion constant $C$ is large?

---

> ### Author Response · Authors · 2025-11-22
>
> Thank you for your helpful comments.
>
> Reply to Q1: In practice, many models are trained/tested on the same set of datasets/benchmarks, so they are aligned to similar ground truth. So it is reasonable to assume that the models we choose approximate each other; in fact, our method relies on this. As the reviewer has mentioned, we computed the ratio D/d for our datasets in Figure 6, and the empirical values are concentrated around 1. On the other hand, using a low-accuracy model as d would probably not help much, as such a model would not align well with the ground truth.
>
>
> Reply to Q2: Our analysis focuses on DiskANN and CoverTree because these are one of the few graph-based algorithms that have suitable theoretical guarantees (though with some modification). And our paper studies how to adapt the existing analysis to our bi-metric case. For other popular graph-based algorithms, such as HNSW and NSG, we agree that they perform very well in practice. However, there is evidence [1] that those algorithms exhibit linear query times for some (synthetic) data sets. However, our method could still yield good empirical performance for those algorithms, without guarantees.
>
> Reply to Q3: The dependence of $\lambda_d$ is necessary for theoretical analysis, as no efficient algorithm can achieve a logarithmic running time when the dimension is not bounded. This has been shown, e.g., in [2]
>
> Reply to Q4: Indeed, we have examined the influence of different initializations in our ablation studies. Please refer to Figure 4 in Appendix D.
>
> Reply to Q5: We chose text retrieval and the BEIR benchmark because it is a widely used benchmark with a comprehensive set of models of different sizes and capacities. We’ll explore other modalities as our future work.
>
> [1] Indyk, Piotr, and Haike Xu. "Worst-case performance of popular approximate nearest neighbor search implementations: Guarantees and limitations." Advances in Neural Information Processing Systems 36 (2023): 66239-66256.
>
> [2] Krauthgamer, Robert, and James R. Lee. "The black-box complexity of nearest-neighbor search." Theoretical Computer Science 348.2-3 (2005): 262-276.

---

### Meta-Review · Area_Chair_XaWr · 2026-01-07

**Summary:**

This paper proposes a simple but efficient method for nearest neighbor search under the assumption that there are two similarity metrics: an accurate but expensive metric and a cheaper surrogate. Under the assumption that the proxy metric approximates the ground-truth metric up to a bounded factor, the proposed algorithm is analyzed theoretically when combined with two NNS methods: DiskANN and Cover Tree.

I really appreciate the theoretical analysis of this paper and agree with the reviewers that it is of value.

There are, however, some concerns regarding the setup and empirical analysis.

First, in most of the experiments, the authors use embedding models (dual encoders) to get both cheap and complex metrics. This makes it potentially possible to pre-compute item embeddings, which is typically done in practice in various production systems. As a more practical scenario, it would be reasonable to consider cross-encoders as complex models. This is partially addressed in the paper by the LLM experiment, but the reviewers observe that the results of this experiment are not very convincing. The authors mentioned that they have done preliminary experiments with cross-encoders, but the results are not discussed in the paper.

Also, this paper assumes that both expensive and cheap metrics are given, which can be considered a limitation. There is an important research direction on how to properly design a cheap metric given an expensive one via, e.g., distillation (usually, the expensive model is a cross-encoder). The assumption that a good enough cheap metric is already given simplifies the problem. I would also like to note that there are relevant papers that consider a similar setup, but with only an expensive metric available. For instance, Yadav et al. “Efficient nearest neighbor search for cross-encoder models using matrix factorization” (EMNLP 2022). In this work (and some follow-up works), it is assumed that there is an expensive metric based on which a cheaper alternative is obtained for faster nearest neighbor search. This line of research is very relevant to the current work (while using slightly different assumptions), and there are also some theoretical guarantees for such approaches obtained in follow-up works. Comparison with such works would improve the current study.

Also, if I am not mistaken, the theoretical analysis is performed under the assumption that at query time only the metric $D$ is used. Thus, theoretical analysis is conducted for an approach similar to the single metric baseline (but with a different graph construction). Also, if I understand correctly, the single metric baseline is similar to the approach discussed in Morozov et al. (2019), but again with a different graph construction. In the experiments, however, first some computations are done using the cheaper metric $d$, and this is shown empirically to give a significant performance boost.

To summarize, while I greatly appreciate the theoretical analysis, at this point, I consider this paper borderline, and I am leaning towards rejection since I believe that a better positioning of this work within the existing literature would improve the paper.

**Reviewer Concerns:**

Some of the reviewers' concerns are:

- Methodological novelty is limited. The authors argue that the simplicity of the approach is its advantage, and I agree with that. (Also, the proposed approach has theoretical guarantees.)

- Some of the experiments use embeddings as a complex model, which can potentially be precomputed. This questions the assumptions of the paper. However, this is not the case for the LLM experiment.

- In the LLM experiment, the method improves over the baseline in only half of the cases.

- There is a suggestion to add experiments with open-source cross-encoders. The authors write that they conducted some preliminary experiments with them, but the results are not discussed.

- Retrieve-then-rerank can be easier combined with sparse search or filtering. I agree with this concern: it is non-trivial to implement filtering in graph-based search based on the expensive metric $D$.

- It is a strong condition that $d$ should be a good approximation of $D$. The authors explained that some assumptions are needed.

- In the experiments, it is shown that $d$ and $D$ are aligned, but this can be the effect of the considered embedders.

- Reproducibility issues since the source code is not available for review.

**Reviewer Scores:**

The initial scores are 8 (JPva), 8 (SDf3), 4 (hixp), 4 (aQro), 2 (GeBN). During the discussion, the score of Reviewer GeBN was increased to 4. I believe that the score of Reviewer aQro would not change after the discussion.

---

### Decision · Program_Chairs · 2026-01-26

Reject